# Systems of Precision: Coherent Probabilities on Pre-Dynkin Systems and Coherent Previsions on Linear Subspaces [note 1]

**DOI:** 10.3390/e25091283

**Published:** 2023-08-31

**Authors:** Rabanus Derr, Robert C. Williamson

**Affiliations:** 1Department of Computer Science, University of Tübingen, 72076 Tübingen, Germany; 2Department of Computer Science, University of Tübingen, Germany and Tübingen AI Center, 72076 Tübingen, Germany; bob.williamson@uni-tuebingen.de

**Keywords:** pre-Dynkin system, Dynkin system, coherence, extendability, quantum probability, intersectionality, prevision, imprecise probability, partial expectation

## Abstract

In the literature on imprecise probability, little attention is paid to the fact that imprecise probabilities are precise on a set of events. We call these sets *systems of precision*. We show that, under mild assumptions, the system of precision of a lower and upper probability form a so-called (pre-)Dynkin system. Interestingly, there are several settings, ranging from machine learning on partial data over frequential probability theory to quantum probability theory and decision making under uncertainty, in which, a priori, the probabilities are only desired to be precise on a specific underlying set system. Here, (pre-)Dynkin systems have been adopted as systems of precision, too. We show that, under extendability conditions, those pre-Dynkin systems equipped with probabilities can be embedded into algebras of sets. Surprisingly, the extendability conditions elaborated in a strand of work in quantum probability are equivalent to coherence from the imprecise probability literature. On this basis, we spell out a lattice duality which relates systems of precision to credal sets of probabilities. We conclude the presentation with a generalization of the framework to expectation-type counterparts of imprecise probabilities. The analogue of pre-Dynkin systems turns out to be (sets of) linear subspaces in the space of bounded, real-valued functions. We introduce partial expectations, natural generalizations of probabilities defined on pre-Dynkin systems. Again, coherence and extendability are equivalent. A related but more general lattice duality preserves the relation between systems of precision and credal sets of probabilities.


*When posing problems in probability calculus, it should be required to indicate for which events the probabilities are assumed to exist.*
—Andreĭ Kolmogorov [1] (p. 52)

## 1. Introduction

Scholarship in imprecise probability largely focuses on the imprecision of probabilities. However, imprecise probability models often lead to *precise* probabilistic statements on certain events or gambles, i.e., bounded, real-valued functions. In this work, we follow a hitherto untaken route investigating the *system of precision*, i.e., the set structure on which an imprecise probability is precise. (We elaborate the exact definition of imprecise probabilities and expectation used here in Section 3 and Section 6). It turns out that (pre-)Dynkin systems (as shown in Appendix B, (pre-)Dynkin systems appear under plenty of names) describe the set of events with precise probabilities (cf. Section 3). This event structure is a neglected object in the literature on imprecise probability. In particular, it constitutes a parametrized choice somewhat “orthogonal” to the standard. Roughly stated, existing approaches to imprecise probability generalize the probability measure $\mu_{\sigma }$ in a classical probability space $\left(\Omega ,F_{\sigma },\mu_{\sigma }\right)$. Following Kolmorogov’s classical setup, Ω is the base set, $F_{\sigma }$ a σ-algebra and $\mu_{\sigma }$ a countably additive probability on $F_{\sigma }$. Approaches to imprecise probability often do not even presuppose an underlying measure space (e.g., [2]). However, they are often linked to finitely additive measure spaces (Ω,F,μ), where μ is a finitely additive probability, and F is an algebra of sets (sometimes called a field). We start by generalizing $F_{\sigma }$ from a σ-algebra to a pre-Dynkin system.

This suggestion is practically motivated: what do the following scenarios have in common?

(a)A machine learning algorithm has access to a restricted subset of attributes. It cannot jointly query all attributes simultaneously. This is called “learning on partial, aggregated information” [3]. The reasons might be manifold: for privacy preservation, “not-missing-at-random” features, restricted data base access for acceleration or multi-measurement data sets.(b)Quantum physical quantities, e.g., location and impulse, are (statistically) incompatible [4].(c)A preference ordering on a set of acts gives rise to precise beliefs on a set of events, whereas this belief is not necessarily precise for intersections of such events [5,6].

In all of these scenarios, there does not exist a precise probability over all attributes and events. Or, there is no such precise probability accessible. Two attributes might each on their own exhibit a precise probabilistic description, while a joint precise probabilistic description does not exist. On a more fundamental level, no intersectability is provided. A precise probabilistic description of two events does *not* imply that the intersection of those events possesses a precise probability. The set system for the description of the events with precise probabilities which independently turned up in the various, previously mentioned fields of research is, again, the (pre-)Dynkin system.

The question of intersectability (or “intersectionality”) is of considerable interest in the social sciences where it is used as a label to describe the problem of the *joint* effect of various individual attributes on social outcomes [7,8,9]. That this notion of intersectionality has something to do with set systems is clear already from the fact that the Venn diagram pictured in Figure 1 is used as an illustration both for the Wikipedia articles on hypergraphs [10] (another name for a set system [11]) and intersectionality [12].

Needless to say, the concept as used in the social sciences is rich, complex, and somewhat vague, which is not necessarily held to be a weakness: “at least part of its success has been attributed to its vagueness” [13] (p. 260). Our interest is in under what circumstances precise probabilities can be ascribed to events; we speculate that such formal results may well contribute to a deeper empirical understanding of social intersectionality, without resorting to fuzzy logic [14] with its potential lack of operational definition as argued for by Cooke [15].

By rethinking the domain of probability measures, one might wonder about the origins of Kolmogorov’s σ-algebra as *the* set system for events which possess probabilities. This links back to the old problem of measurability [16] (pp. 1–5). The measurability problem is the mathematical problem to assign a uniform measure to all subsets of a continuum. Giuseppe Vitali showed in 1905 that this problem is not solvable for countably additive measures [16] (p. 5) (from [17]). Hence, more restricted set systems such as the σ-algebra arose. Isaacs et al. [18] reconsidered this century-old discussion to argue for rationality of imprecise probabilities. We take their argument even further. Inside the borders of mathematical measurability, the set of events which ought to be assigned probabilities is a modeling choice. Measurability is a modeling tool. We show that it is naturally parametrized by the set of (pre-)Dynkin systems.

All of the preceding considerations bring us to the main question of this paper: *What is the system of precision and how does it relate to an imprecise probability on “all” events?* We approach this question from three perspectives.

First, we show that, under mild assumptions, a pair of lower and upper probabilities assign precise probabilities, i.e., lower and upper probability coincide, to events which form a pre-Dynkin system or even a Dynkin system.Second, we define probabilities on pre-Dynkin systems in accordance with the literature on quantum probability, in particular [19]. We argue that probabilities on pre-Dynkin systems, as well as their inner and outer extension, exhibit few desirable properties, e.g., subadditivity cannot be guaranteed. Hence, extendability, the ability to extend a probability from a pre-Dynkin system to a larger set structure, turns out to be crucial, as it implies coherence of the probability defined on the pre-Dynkin system. This observation links together the research from probabilities defined on weak set structures [6,19,20] to imprecise probabilities [2,21]. Furthermore, extendability guarantees the existence of a nicely behaving, so-called coherent extension. We finally show that the inner and outer extension of a probability defined on a pre-Dynkin system is always more pessimistic than its corresponding lower and upper coherent extension.Last, we develop a duality theory between pre-Dynkin systems on a predefined base measure space and their respective credal sets of probabilities. The credal sets consist of all probabilities which coincide with the pre-defined measure on a pre-Dynkin system. A so-called Galois connection links together the containment structure on the set of set systems with the containment structure on the set of credal sets.

We conclude our presentation with a generalization to expectation-type counterparts of imprecise probabilities in Section 6. These are often called previsions, e.g., in [2]. Our main question thus generalizes to: *What is the system of precision and how does it relate to an imprecise expectation on “all” gambles?* In this case, by “system of precision” we mean the set of gambles on which a lower and upper expectation coincide.

First, we propose a generalization of a finitely additive probability defined on a pre-Dynkin system. More concretely, we define *partial expectations* which correspond to expectation functionals which are only defined on a set of linear subspaces of the space of all gambles. However, on those linear subspaces, they behave like “classical” (finitely additive) expectations.Second, we show that under some properties, imprecise expectations are precise on a linear subspace of the linear space of gambles. (cf. Section 3)Third, we present a natural generalization of extendability for partial expectations, which again turns out to be equivalent to coherence of the partial expectation.Last, analogous to the lattice duality (A lattice is a poset with pairwise existing minimum and maximum. The duality is expressed via an antitone lattice isomorphism.) described in Section 5, we present a lattice duality for linear subsets of the space of gambles and credal sets which define coherent lower and upper previsions.

In summary, our work makes contributions in-between the research field of imprecise probabilities, probabilities defined on general set structures, and partially defined expectation functionals. Part of this work has been presented at the International Symposium on Imprecise Probabilities: Theories and Applications under the title “The Set Structure of Precision” [22]. The following version is a more complete and exhaustive presentation of this conference version. We included all omitted proofs of the conference version. We elaborated the content of Section 5. We added an entire section about the generalization to expectation-type counterparts of probabilities on pre-Dynkin systems (Section 6). We presented relations to different research areas in more detail. We shortly discussed countably additive probabilities on Dynkin systems in the appendix, as we put emphasis on finitely additive probabilities in the main text. Before we begin the structural investigation of pre-Dynkin systems, we first introduce the used notation and fix the mathematical framework.

### Notation and Technical Details

As we deal with a lot of sets, sets of sets, and rarely even sets of sets of sets in this paper, we agree on the following notation: sets are written with capital Latin or Greek letter, e.g., *A* or Ω. Sets of sets are denoted A. Sets of sets of sets obtain the notation A. As usual, R is reserved for the set of real numbers, N for the natural numbers. The power set of a set *A* is written as $2^{A}$.

In the course of this work, we require the notions of σ-algebras and algebras (of sets). An algebra is a subset of $2^{\Omega }$ which contains the empty set and is closed under complement and finite union. A σ-algebra is an algebra which is closed under countable union [23] (Definition 1.1). ( Our notion of an algebra should not be confused with the notion of an algebra over a field.) (Probability) measures are denoted by lowercase Greek letters, e.g., μ, ν and ψ, except for σ. Generally, we use “σ” to emphasize the countable nature of a mathematical object. This becomes clear when we define Dynkin systems (Definition 1). Other functions are denoted by lowercase Latin letters, e.g., *f* and *g*.

Regarding the technical setting, we roughly follow the setup of [2] (§3.6 and Appendix D). For a summary, see Table 1.

Let Ω be an arbitrary set. In several examples Ω=[n], where [n] denotes the set {1,…,n}. The set B(Ω) is defined as the set of all real-valued, bounded functions on Ω. We call those functions *gambles*. For instance, $\chi_{A}$, the indicator function of A⊆Ω, is in B(Ω). The supremum norm, ${∥f∥}_{\sup}:=\sup_{\omega \in \Omega }\left|f\left(\omega \right)\right|$ makes B(Ω) a topological linear vector space [24]. With ba(Ω), we denote the set of all bounded, signed, finitely additive measures on $2^{\Omega }$. In fact, ba(Ω) is the topological dual space of B(Ω). So, in particular, every continuous linear functional $\nu \in B{\left(\Omega \right)}^{*}$ can be identified with a bounded, signed, finitely additive measure [24]. (As the linear functional is defined on a normed space, continuity and boundedness are equivalent.) For this reason, we use, with minor abuse of notation, the same notation for bounded, signed, finitely additive measures and for continuous linear functionals in the dual space of B(Ω), i.e., we write ν(f)=∫fdν. The dual space ba(Ω) gets equipped with the weak${}^{★}$ topology, i.e., the weakest topology which makes all evaluation functionals of the form $f^{*}\in \mathrm{ba}{\left(\Omega \right)}^{*}$ such that $f^{*}\left(\nu \right):=\int fd\nu$ for some f∈B(Ω) continuous (for more details see, e.g., [25] (p. 758)). With Δ⊆ba(Ω), we denote the convex, weak${}^{★}$-closed subset of finitely additive probability measures. The set Δ plays a major role in Walley’s theory of previsions, as the measures in Δ are in one-to-one correspondence to his linear previsions [2] (Theorem 3.2.2). The operator $\bar{\mathrm{co}}\;$ is the convex, weak${}^{★}$ closure on the space ba(Ω). We further introduce the following two notations: let $F\subseteq 2^{\Omega }$ be an algebra. Then, S(Ω,F)⊆B(Ω) denotes the linear subspace of simple functions on F, i.e., scaled and added indicator functions of a finite number of disjoint sets (cf. [26] (Definition 4.2.12)). Let $F_{\sigma }\subseteq 2^{\Omega }$ be a σ-algebra. Then, $B\left(\Omega ,F_{\sigma }\right)\subseteq B\left(\Omega \right)$ denotes the linear subspace of all bounded, real-valued, $F_{\sigma }$-measurable functions. Equipped with these notions and tools we are ready for a first preliminary question.

## 2. What Is a (Pre-)Dynkin System?

In this work, the main objects under consideration are pre-Dynkin systems and Dynkin systems. A (pre-)Dynkin system is a set system on Ω. It contains the empty set, is closed under complement and (countable) disjoint union. More formally:

**Definition 1** ((Pre-)Dynkin system)**.***We say $D\subseteq 2^{\Omega }$ is a* pre-Dynkin system *on some set *Ω* if and only if all of the following conditions hold:**(a)* *∅∈D,**(b)* *D∈D implies $D^{c}:=\Omega \setminus D\in D$**(c)* *C,D∈D with C∩D=∅ implies C∪D∈D.**We call $D_{\sigma }\subseteq 2^{\Omega }$ a* Dynkin system *if and only if the conditions (a), (b) and*
*(c’)* 
*let ${\left\{D_{i}\right\}}_{i\in N}\subseteq D_{\sigma }$, if for all i,j∈N with i≠j it holds $D_{i}\cap D_{j}=\emptyset$ then ${⋃}_{i\in N}D_{i}\in D_{\sigma }$,*

*are fulfilled.*


Observe that every Dynkin system is a pre-Dynkin system. We will denote pre-Dynkin systems by the use of D, in contrast to $D_{\sigma }$ for Dynkin systems. This should not be confused with D(A) for $A\subseteq 2^{\Omega }$, which is the intersection of all pre-Dynkin systems which contain A, i.e., the smallest pre-Dynkin system containing A. (For A=∅ we define D(A)={∅,Ω}.) In other words, D(A) is the *pre-Dynkin hull* generated by A. The following short lemma will be helpful in later proofs.

**Lemma 1** (Closedness under Set Difference)**.**
*Let $D\subseteq 2^{\Omega }$ be a pre-Dynkin system, if A,B∈D and A⊆B, then B∖A∈D.*


**Proof.** Let A,B∈D and A⊆B. Then, ${\left(B\setminus A\right)}^{c}=B^{c}\cup A$. Since D is closed under complement and disjoint union $B^{c}\cup A\in D$. But, then again, the complement, B∖A, is in D. □

In classical probability theory, Dynkin systems appear as a technical object required for the measure-theoretic link between cumulative distribution functions and probability measures (cf. [23] (Proof of Lemma 1.6)). In particular, every σ-algebra, the well-known domain of probability measures, is a Dynkin system. Thus, all statements within this work are generalizations of classical probability theoretical results. We give a short example of a pre-Dynkin system, which is not an algebra in the following. This example gets reused to illustrate forthcoming statements.

**Example** **1.**
*The smallest pre-Dynkin system, up to isomorphisms, which is not an algebra can be defined on $\Omega_{4}:=\left\{1,2,3,4\right\}$. It is given by $D_{4}:=\left\{\emptyset ,12,34,13,24,\Omega_{4}\right\}$, where we write 12 as a shorthand for {1,2}.*


Pre-Dynkin and Dynkin systems naturally arise in probability theory. For instance, the set of all subsets A⊆N, such that the natural density $\mu \left(A\right)=\lim_{n\to \infty }\frac{\left|A\cap [n]\right|}{n}$ exists is a pre-Dynkin system $D_{N}$ but not an algebra [20]. It is sometimes called the density logic [27] and constitutes the foundation of von Mises’ century-old frequential theory of probability [28] (refined and summarized in [29]). Intriguingly, this was used as an example by Kolmogorov [1] of a measure defined on a restricted set system for which it is desired to extend the measure to the power set $2^{N}$ (cf. § Section 4); see the discussion in [30] (pp. 11–14), who observed (page 14) that “the main problem is *non-uniqueness* of an extension” and that such extended measures are impossible to verify from observed frequencies, because the relative frequencies do not converge for events in $2^{N}\setminus D_{N}$. The non-uniqueness is naturally handled in the present paper by working with lower and upper previsions (or lower and upper probabilities) (cf. [31]).

Another class of Dynkin systems occurs in so-called marginal scenarios [32]. Marginal scenarios are settings in which marginal probability distributions for a subset of a set of random variables are given, but not the entire joint distribution. This restricted “joint measurability” of the involved random variables can be expressed via Dynkin systems [33] (Example 4.2), [34].

Pre-Dynkin systems are so helpful because they structurally align with finitely additive probability measures. The same statement holds for Dynkin systems and countably additive probabilities. If we know the probability of an event, then we know the probability of the complement; i.e., the event does not happen. If we know the probability of several events which are disjoint, then we know the probability of the union, which is just the sum. Probabilities following their standard definition go hand in hand with Dynkin systems. We see this observation manifested in many following statements.

Remarkably, (pre-)Dynkin systems appeared under a variety of names (cf. Appendix B). Fundamental to all its regular, independent occurences in many research areas is the need for a set structure which does not allow for arbitrary intersections.

### 2.1. Compatibility

(Pre-)Dynkin systems are not necessarily closed under intersections. However, when the intersection of two sets (events) is contained in the (pre-)Dynkin system, we call the two events *compatible*.

**Definition 2** (Compatibility)**.***Let A,B be elements in a pre-Dynkin system D, then A and B are* compatible *if and only if A∩B∈D.*

This definition follows the definitions given in, e.g., [19,33,35]. (It should not be confused with the very similar, and sometimes equivalent, notion of commutativity in logical structures [36] (Definition 14) (cf. Appendix E).) Compatibility in pre-Dynkin systems is a symmetric relation, but it is not necessarily transitive. Furthermore, it is complement inherited, i.e., if A,B are compatible in a pre-Dynkin system, then so are $A,B^{c}$ [4] (Lemma 3.6). Lastly, compatibility, even though expressed as intersectability, i.e., “closed under intersection”, can be equivalently expressed as unifiability, i.e., “closed under union”.

**Lemma 2** (Cup gives Cap gives Cup)**.**
*Let D be a pre-Dynkin system and A,B∈D. Then*

$$\begin{array}{c} A\cap B\in D\Leftrightarrow A\cup B\in D \end{array}$$



**Proof.** Using Lemma 1 for pre-Dynkin systems, we can quickly see that the following two decompositions give the desired equivalence:
For the “⇒”-direction: A∪B=(A∖(A∩B))∪B. The fact A,A∩B,B∈D implies (A∖(A∩B))∪B∈D.For the “⇐”-direction: A∩B=A∖((A∪B)∖B). The fact A,A∪B,B∈D implies A∖((A∪B)∖B)∈D. (A related result for Dynkin systems is given in [19] (5.1).) □


**Example** **2.**
*We reconsider the set $\Omega_{4}$ and pre-Dynkin system $D_{4}$ from Example 1. The elements 12 and 34 are intersectable $12\cap 34=\emptyset \in D_{4}$ and unifiable $12\cup 34=\Omega_{4}\in D_{4}$. The elements 12 and 13 are not intersectable $12\cap 13=1\notin D_{4}$ and not unifiable $12\cup 13=123\notin D_{4}$.*


The term “compatibility” underlines that closedness under intersection gets loaded with further meaning in the context of probability theory. As we define in the next section, D is the set of “measurable” events, i.e., events which get assigned a probability. Hence, two events, A,B, are called compatible if and only if a precise joint probabilistic description, i.e., a precise probability of A∩B, exists. For a more thorough discussion of the nature of compatibility (and its cousin commutativity) we point to the literature on quantum probability, e.g., [37] (Definition 3.12), or [38].

Compatibility is not only a property of elements in pre-Dynkin systems. One can take compatibility as a primary notion; i.e., one requires the statements of Lemma 2 and [4] (Lemma 3.6) to hold. Then, a set structure which contains the empty set and the entire base set and is equipped with this notion of compatibility is a pre-Dynkin system [30] (Definition 5.1). (It is called semi-algebra in [30] (Definition 5.1).)

Interestingly, the assumption of arbitrary compatibility is fundamental to most parts of probability theory. σ-algebras, the domain of probability measures, are exactly those Dynkin systems in which all events are compatible with all others [35] (Theorem 2.1). Algebras are exactly those pre-Dynkin systems in which all events are compatible with all others. Surprisingly, it turns out that, as well, all pre-Dynkin systems can be dissected into such “blocks” of full compatibility. Every pre-Dynkin system consists of a set of maximal algebras, which we call *blocks*. In particular, maximality here stands for the following: there is no algebra contained in D such that some $A_{i}$ is a strict sub-algebra of this algebra. Similar and related results can be found in [39,40,41,42].

**Theorem** **1**(Pre-Dynkin Systems Are Made Out of Algebras). *Let D be a pre-Dynkin system on Ω. Then, there is a unique family of maximal algebras ${\left\{A_{i}\right\}}_{i\in I}$ such that $D={⋃}_{i\in I}A_{i}$. We call these algebras the* blocks *of D.*

**Proof.** For the proof, we require the definition of a compatible subset of D. A subset A⊆D is compatible if all elements are completely compatible, i.e., any finite intersection of elements in A is contained in D. This is indeed a stronger requirement than pairwise compatibility (cf. Definition 2). Certainly, every subset A⊆D is compatible if and only if every finite subset of A is compatible. Hence, compatibility is a property of so-called finite character [25] (Definition 3.46). Then, Tuckey’s lemma (e.g., [25] (Theorem 6.20.AC5)) guarantees that any compatible subset of D is contained in a maximal compatible subset. Since every element D∈D is in at least one compatible subset, e.g. $\{\emptyset ,D,D^{c},\Omega \}\subseteq D$, the (unique) set of maximal compatible subsets ${\left\{A_{i}\right\}}_{i\in I}$ covers the entire pre-Dynkin system D. It remains to show that the maximal compatible subsets are algebras. Consider a maximal compatible subset $A_{i}$. First, $\emptyset \in A_{i}$ as ∅ is compatible to all sets in $2^{\Omega }$. Second, $A_{i}$ is closed under finite intersection, otherwise there would exist a finite combination of elements $A_{1},\ldots ,A_{n}\subseteq A_{i}$ such that $A_{\cap }:={⋂}_{j=1}^{n}A_{j}\in D$, but $A_{\cap }\notin A_{i}$. Then, one can easily see that $A_{i}\cup \left\{A_{\cap }\right\}$ would be a compatible subset which strictly contained $A_{i}$. This is impossible, since $A_{i}$ is maximal. Finally, $A_{i}$ is closed under complement. Consider $A\in A_{i}$, we show that $A_{i}\cup \left\{A^{c}\right\}$ is again a compatible subset. Let $A_{1},\ldots ,A_{n}\subseteq A_{i}$ be an arbitrary finite collection of subsets, then $A\cap {⋂}_{j=1}^{n}A_{n}\in D$, hence $A^{c}\cap {⋂}_{j=1}^{n}A_{n}\in D$[4] (Lemma 3.6). By maximality of $A_{i}$, we then know $A^{c}\in A_{i}$. □

**Example** **3.**
*The pre-Dynkin system $D_{4}$ of Example 1 consists of the algebras $\left\{\emptyset ,12,34,\Omega_{4}\right\}$ and $\left\{\emptyset ,13,24,\Omega_{4}\right\}$.*


Theorem 1 simplifies several follow-up observations. Instead of pre-Dynkin systems, we can equivalently consider a set of algebras. However, not every union of algebras is a pre-Dynkin system. If these algebras form a compatibility structure, i.e., a set of maximal π-systems, which are non-empty set systems closed under finite intersections, then their union is a pre-Dynkin system (Definition A2 and Theorem A1 in Appendix A). Analogous results for Dynkin systems and σ-algebras exist and are given in Section D.1. In summary, pre-Dynkin systems are set structures which do not allow for arbitrary intersections but can be split into maximal intersectable subsets, their *blocks*.

### 2.2. Probabilities on Pre-Dynkin Systems

We require a notion of probability on a pre-Dynkin system to elaborate the relationship of imprecise probability and the system of precision in the following. Probabilities are classically defined on σ-algebras. We generalize this definition as e.g., stated in [23] (p. 18f) to pre-Dynkin systems.

**Definition 3** (Probability Measure on a Pre-Dynkin System)**.***Let D be a pre-Dynkin system. We call a function μ:D→[0,1] a* finitely additive probability measure on D *if and only if it fulfills the following two conditions:**(a)* Normalization*: μ(∅)=0 and μ(Ω)=1.**(b)* Additivity*: let A,B∈D and A∩B=∅. Then, μ(A∪B)=μ(A)+μ(B).**If condition (b) can be extended to countable subsets of D, then we call μ*countably additive probability measure on D*: let I⊆N and ${\left\{A_{i}\right\}}_{i\in I}$ such that $A_{i}\in D$ for all i∈I and $A_{i}\cap A_{j}=\emptyset$ for i≠j, i,j∈I and ${⋃}_{i\in I}A_{i}\in D$. Then $\mu \left({⋃}_{i\in I}A_{i}\right)=\sum_{i\in I}\mu \left(A_{i}\right)$.*

For the sake of readability, we use “probability” and “probability measure” exchangeably. Probabilities on pre-Dynkin systems are monotone, i.e., for A,B∈D, if A⊆B, then μ(A)≤μ(B). This can be seen when applying Lemma 1 and Definition 3. But, in contrast to a probability defined on a σ-algebra, a probability on a pre-Dynkin system is not necessarily *modular*, i.e., for A,B∈D, μ(A)+μ(B)=μ(A∪B)+μ(A∩B) [43] (p. 16). (It is, however, possible to define modular probabilities on pre-Dynkin systems. This leads to a fixed parametrization of probability functions already on simple examples [44] (p. 125).) It is that sophisticated interplay of set structure and probability function which leads us through this paper. In particular, why should we consider pre-Dynkin systems?

## 3. Imprecise Probabilities Are Precise on a Pre-Dynkin System

As we now demonstrate, pre-Dynkin systems are, under mild assumptions, the systems of precision. To make this formal, we solely require a normed, conjugate pair of lower and upper probability which fulfill super (resp. sub)-additivity and possibly a continuity assumption.

**Theorem 2** (Imprecise Probability Induces a (Pre-)Dynkin System)**.**
*Let $\ell :2^{\Omega }\to \left[0,1\right]$ and $u:2^{\Omega }\to \left[0,1\right]$ be two set functions, for which all the following properties hold:*

*(a)* 
*Normalization: u(∅)=ℓ(∅)=0.*
*(b)* 
*Conjugacy: $u\left(A\right)=1-\ell \left(A^{c}\right)$ for $A,A^{c}\in 2^{\Omega }$.*
*(c)* 
*Subadditivity of u: for $A,B\in 2^{\Omega }$ such that A∩B=∅, then u(A∪B)≤u(A)+u(B).*
*(d)* 
*Superadditivity of ℓ: for $A,B\in 2^{\Omega }$ such that A∩B=∅, then ℓ(A∪B)≥ℓ(A)+ℓ(B).*

*Then u and ℓ define a finitely additive probability measure ${\mu :=u|}_{D}{=\ell |}_{D}$ on the pre-Dynkin system $D:=\left\{A\in 2^{\Omega }:\ell \left(A\right)=u\left(A\right)\right\}\subseteq 2^{\Omega }$. If either u fulfills*

*(e)* 
*Continuity from below: for $A_{n}\in 2^{\Omega }$ with $A_{n}\subseteq A_{n+1}$ such that ${⋃}_{n=1}^{\infty }A_{n}=A\in 2^{\Omega }$, then*

*$\lim_{n\to \infty }u\left(A_{n}\right)=u\left(A\right)$,*

*or ℓ fulfills*

*(e’)* 
*Continuity from above: for $A_{n}\in 2^{\Omega }$ with $A_{n+1}\subseteq A_{n}$ such that ${⋂}_{n=1}^{\infty }A_{n}=A\in 2^{\Omega }$, then*

*$\lim_{n\to \infty }\ell \left(A_{n}\right)=\ell \left(A\right)$,*

*then u and ℓ define a countably additive probability measure $\mu_{\sigma }:=u{{|}_{D_{\sigma }}=\ell |}_{D_{\sigma }}$ on the Dynkin system $D_{\sigma }:=\left\{A\in 2^{\Omega }:\ell \left(A\right)=u\left(A\right)\right\}\subseteq 2^{\Omega }$.*


**Proof.** We start proving the first part of the theorem. Let
(1)$$D:=\{A\in 2^{\Omega }:\ell \left(A\right)=u\left(A\right)\}.$$We show that D is a pre-Dynkin system. First, ∅∈D by assumption (a). Second, let D∈D. Then, $u\left(D^{c}\right)=1-\ell \left(D\right)=1-u\left(D\right)=\ell \left(D^{c}\right)$ by the conjugacy relation. Third, let ${\left\{A_{i}\right\}}_{i\in I}\subseteq D$ for finite I⊆N such that $A_{i}\cap A_{j}=\emptyset$ for all i≠j, then
$$\begin{array}{cc} \sum_{i\in I}\ell \left(A_{i}\right) & \overset{\left(d\right)}{\leq }\ell \left(\underset{i\in I}{⋃}A_{i}\right)\\ \; & \overset{\left(★\right)}{\leq }u\left(\underset{i\in I}{⋃}A_{i}\right)\\ \; & \overset{\left(c\right)}{\leq }\sum_{i\in I}u\left(A_{i}\right)\\ \; & \overset{Equation\;(1)}{=}\sum_{i\in I}\ell \left(A_{i}\right). \end{array}$$For (★), observe that ℓ(A)≤u(A) for all $A\in 2^{\Omega }$, since
$$\begin{array}{cc} \ell \left(A\right)+\ell \left(A^{c}\right) & \leq \ell \left(A\cup A^{c}\right)=1=u\left(A\cup A^{c}\right)\leq u\left(A\right)+u\left(A^{c}\right), \end{array}$$
and thus,
$$\begin{array}{cc} \ell \left(A\right)+\ell \left(A^{c}\right) & \leq u\left(A\right)+u\left(A^{c}\right)\\ \Leftrightarrow \ell (A)+1-u(A) & \leq u(A)+1-\ell (A)\\ \Leftrightarrow \ell (A) & \leq u(A). \end{array}$$Concluding, we define ${\mu :=\ell |}_{D}{=u|}_{D}$ for which it is trivial to show that it is a finitely additive probability on D.For the second part, we first notice that continuity from below and from above are equivalent for conjugate set functions on set systems which are closed under complement [43] (Proposition 2.3). Next, we show that subadditivity of *u* and continuity from below (of *u*) imply σ-subadditivity of *u*: for ${\left\{A_{i}\right\}}_{i\in I}\subseteq 2^{\Omega }$ such that I⊆N and $A_{i}\cap A_{j}=\emptyset$ for all i≠j with i,j∈I then $u\left({⋃}_{i\in I}A_{i}\right)\leq \sum_{i\in I}u\left(A_{i}\right)$. In the case that *I* is finite, subadditivity of *u* is provided by assumption. For infinite *I*, we can construct an increasing sequence of sets, namely $B_{j}={⋃}_{1\leq i\leq j}A_{i}$, so that $B_{j}\subseteq B_{j+1}$. Furthermore, ${⋃}_{j=1}^{\infty }B_{j}={⋃}_{i\in I}A_{i}$. Thus,
$$\begin{array}{cc} u\left(\underset{i\in I}{⋃}A_{i}\right) & =u\left(\overset{\infty }{\underset{j=1}{⋃}}B_{j}\right)\\ \; & \overset{\left(e\right)}{=}\lim_{j\to \infty }u\left(B_{j}\right)\\ \; & =\lim_{j\to \infty }u\left(\underset{1\leq i\leq j}{⋃}A_{i}\right)\\ \; & \overset{\left(d\right)}{\leq }\lim_{j\to \infty }\sum_{1\leq i\leq j}u\left(A_{i}\right)\\ \; & =\sum_{i\in I}u\left(A_{i}\right). \end{array}$$The same argument holds analogously for superadditivity and continuity from above of *ℓ* which is implied by continuity from below and the conjugacy relationship [43] (Proposition 2.3). In summary, the proof of the first part can then be applied again, now without the restriction that I⊆N is finite. Instead, it potentially is countable. □

**Example** **4.**
*Remember, $\Omega_{4}=\left\{1,2,3,4\right\}$. We define $\ell :2^{\Omega_{4}}\to \left[0,1\right]$ by ℓ(1)=0, ℓ(2)=0.3, ℓ(3)=0, ℓ(4)=0.3, ℓ(12)=0.5, ℓ(34)=0.5, ℓ(13)=0.2, ℓ(24)=0.8, ℓ(14)=0.3, ℓ(23)=0.3, ℓ(123)=0.5, ℓ(124)=0.8, ℓ(134)=0.5, ℓ(234)=0.8, $\ell (\Omega_{4})=1$ and $u:2^{\Omega_{4}}\to \left[0,1\right]$ by $u\left(A\right)=1-\ell \left(A^{c}\right)$. It is easy to show that ℓ and u fulfill the assumptions (a), (b), (c) and (d) in Theorem 2. The imprecise probabilities u and ℓ coincide on $\left\{\emptyset ,12,34,13,24,\Omega_{4}\right\}$, the pre-Dynkin system described in Example 1. The example is illustrated in Figure 2. In this figure $\ell ={\underline{\mu }}_{D_{4}}$ and $u={\bar{\mu }}_{D_{4}}$.*


In summary, imprecise probabilities are, under mild assumptions, precise on a pre-Dynkin system or even a Dynkin system. (The events in the system of precisions are called *unambiguous events* in the decision theory literature [5].) This, importantly, is also the case if the system of precision is strictly larger than the trivial pre-Dynkin systems {∅,Ω}. Exemplarily, a pair of conjugate, coherent lower and upper probability (e.g., [2] (§2.7.4)) fulfills the conditions (a)–(d). However, in several cases (e.g., distorted probability distributions), imprecise probabilities are just precise on the *system of certainty*, i.e., the events which possess 0 or 1 probability (Proposition A1). Concluding, the system of precision is a pre-Dynkin system $D\subseteq 2^{\Omega }$. What if we first define precise, finitely additive probabilities on a pre-Dynkin system, i.e., we fix a system of precision? We can then ask for “imprecise probabilities” deduced from this probability which are defined on a larger set structure, e.g., an algebra in which the pre-Dynkin system is contained.

## 4. Extending Probabilities on Pre-Dynkin Systems

Precise probabilities on pre-Dynkin system naturally arise in many, distinct, applied scenarios as we argued in the Introduction (Section 1). However, we acknowledge that the definition of probabilities on pre-Dynkin systems is mathematically cumbersome. The possibilities to prove standard theorems is very limited as the approaches by Gudder [4,35], Gudder and Zerbe [45] demonstrate. However, if we consider a probability defined on a pre-Dynkin system as an imprecise probability on a larger set system with a fixed system of precision, we possibly obtain a richer, mathematical toolkit to work with. In this case the larger set system preferably is an algebra in which the pre-Dynkin system is contained. It remains to clarify how we construct the imprecise probability from the precise probability on the pre-Dynkin sytem.

### 4.1. Inner and Outer Extension

A simple but, as we show, unsatisfying solution is the use of an inner and outer measure extension. It does not rely on imposing any conditions on the probability defined on the pre-Dynkin system. We pay for this generality with the few properties that we can derive for the obtained extension.

**Proposition** **1** (Inner and Outer Extension [6] (Lemma 2.2))**.**
*Let D be a pre-Dynkin system on Ω and μ a finitely additive probability measure on D. The* inner probability measure
$$\begin{array}{cc} \mu_{*}\left(A\right) & :=\sup\left\{\mu \left(B\right):A⊇B\in D\right\},\;\forall A\in 2^{\Omega }, \end{array}$$*and* outer probability measure
$$\begin{array}{cc} \mu^{*}\left(A\right) & :=\inf\left\{\mu \left(B\right):A\subseteq B\in D\right\},\;\forall A\in 2^{\Omega }, \end{array}$$*define $\mu_{*},\mu^{*}:2^{\Omega }\to \left[0,1\right]$, i.e., all of the following conditions are fulfilled:*
*(a)* 
*Normalization: $\mu^{*}\left(\emptyset \right)=0$, $\mu_{*}\left(\Omega \right)=1$.*
*(b)* 
*Conjugacy: $\mu^{*}\left(A\right)=1-\mu_{*}\left(A^{c}\right),\;\forall A\in 2^{\Omega }$.*
*(c)* 
*Monotonicity: for $A,B\in 2^{\Omega }$, if A⊆B, then $\mu^{*}\left(A\right)\leq \mu^{*}\left(B\right)$.*

*Furthermore, $\mu_{*}$ is superadditive, for $A,B\in 2^{\Omega }$ if A∩B=∅, then $\mu_{*}\left(A\cup B\right)\geq \mu_{*}\left(A\right)+\mu_{*}\left(B\right)$. But $\mu^{*}$ is not generally subadditive.*


**Example** **5.**
*For $D_{4}$, as in Example 4, let $\mu :D_{4}\to \left[0,1\right]$ be defined as μ(∅)=0,μ(12)=0.5, μ(34)=0.5,μ(13)=0.2,μ(24)=0.8,μ(Ω)=1. The inner and outer extension of μ on $D_{4}$ is $\mu_{*}\left(\emptyset \right)=0,\mu_{*}\left(1\right)=\mu_{*}\left(2\right)=\mu_{*}\left(3\right)=\mu_{*}\left(4\right)=0,\mu_{*}\left(12\right)=0.5,\mu_{*}\left(34\right)=0.5$, $\mu_{*}\left(13\right)=0.2,\mu_{*}\left(24\right)=0.8,\mu_{*}\left(14\right)=0,\mu_{*}\left(23\right)=0,\mu_{*}\left(123\right)=0.5,\mu_{*}\left(124\right)=0.8$, $\mu_{*}\left(134\right)=0.5,\mu_{*}\left(234\right)=0.8,\mu_{*}\left(\Omega_{4}\right)=1$ and $\mu^{*}=1-\mu_{*}$. The inner and outer extension are not coherent (Definition 5). In particular, the outer extension is not subadditive: $\mu^{*}\left(14\right)=1-\mu_{*}\left(23\right)=1>0.2+0.5=\left(1-\mu_{*}\left(234\right)\right)+\left(1-\mu_{*}\left(123\right)\right)=\mu^{*}\left(1\right)+\mu^{*}\left(4\right)$. The example is illustrated in Figure 2.*


In conclusion, the inner and outer extension provides an imprecise probability, which is not necessarily coherent (cf. Definition 5) and it does not fulfill the conditions required for Theorem 2 to post hoc guarantee that the set of precision is a pre-Dynkin system. We remark that there exist normalized, conjugate, monotone superadditive but not subadditive pairs of probabilities, hence possibly inner and outer probabilities as defined here, whose system of precision is not a pre-Dynkin system (see Example 6). For this reason we now explore another, more powerful extension method.

**Example** **6.**
*Let $\Omega_{3}:=\left\{1,2,3\right\}$. The probability pair defined by $\underline{\nu }\left(\emptyset \right)=\bar{\nu }\left(\emptyset \right)=0$, $\underline{\nu }\left(1\right)=\bar{\nu }\left(1\right)=0.2$, $\underline{\nu }\left(2\right)=\bar{\nu }\left(2\right)=0.2$ and $\underline{\nu }\left(3\right)=0,\bar{\nu }\left(3\right)=0.6$ is precise on $\left\{\emptyset ,1,2,23,13,\Omega_{3}\right\}$, which is obviously not a pre-Dynkin system.*


### 4.2. Extendability and Its Equivalence to Coherence

In the following, we try to entirely embed pre-Dynkin systems equipped with a probability into larger algebras. Then, we extend the probability defined on the pre-Dynkin system in all possible ways to probabilities on the algebra. It turns out that this embedding is only possible under certain conditions on the probability defined on the pre-Dynkin system. We call this condition *extendability*. For the sake of generality, we focus on the extension of finitely additive probabilities from pre-Dynkin systems to algebras here. We treat countably additive probabilities, Dynkin systems and σ-algebras in Appendix D. In addition, all results until Section 4.3 can be formulated in more general terms for non-structured set systems. For the sake of simplicity, we remain within the setting of probabilities defined on pre-Dynkin systems in this work.

Extendability is the property that a probability measure defined on a pre-Dynkin system can be extended to a probability measure on an algebra containing the pre-Dynkin system. Formally:

**Definition 4** (Extendability)**.***Let D be a pre-Dynkin system on Ω. We call a finitely additive probability measure μ on D* extendable *to $2^{\Omega }$ if and only if there is a finitely additive probability measure $\nu :2^{\Omega }\to \left[0,1\right]$ such that ${\nu |}_{D}=\mu$.*

We defined extendability with respect to the power set $2^{\Omega }$. In fact, any relativization to an arbitrary sub-algebra of $2^{\Omega }$ is equivalent. A finitely additive probability defined on D is extendable to any sub-algebra of $2^{\Omega }$ which contains D if and only if it is extendable to $2^{\Omega }$ [26] (Theorem 3.4.4).

The definition is non-vacuous [33,46]. For instance, a probability measure on a pre-Dynkin System is not generally extendable to a measure on the generated algebra (e.g., Example 3.1 in [33]). If a probability is extendable, its extension is in general non-unique.

Extendability of probabilities on (pre-)Dynkin systems has already been part of discussions in quantum probability since 1969 [19] up to more current times [47]. It as well has been of interest in frequential probability theory [20] (Theorem 2). Several necessary and/or sufficient conditions on the structure of D and/or the values of μ are known [33,46,47]. We present here a sufficient and necessary condition discovered by Horn and Tarski [48] and restated in [26] (Theorem 3.2.10). In fact, Theorem 3 can be stated for a more general definition of probabilities on arbitrary set systems e.g., [26] (Theorem 3.2.10), [49] (Proposition 2.2). For the sake of simplicity, we restrict this result to pre-Dynkin systems and probabilities defined on pre-Dynkin systems.

**Theorem** **3**(Extendability Condition [26] (Theorem 3.2.10)). *Let D be a pre-Dynkin system on Ω. A finitely additive probability measure μ on D is extendable to $2^{\Omega }$ if and only if*
(2)$$\begin{array}{c} \sum_{k=1}^{m}\chi_{B_{k}}\left(\omega \right)-\sum_{j=1}^{n}\chi_{A_{j}}\left(\omega \right)\geq 0,\;\forall \omega \in \Omega \Rightarrow \sum_{k=1}^{m}\mu \left(B_{k}\right)-\sum_{j=1}^{n}\mu \left(A_{j}\right)\geq 0 \end{array}$$*for all finite families of sets in D: $A_{1},\ldots ,A_{n},B_{1},\ldots ,B_{m}\in D$.*

**Example** **7.**
*For $D_{4}$ as in Example 4 let $\mu :D_{4}\to \left[0,1\right]$ be defined as μ(∅)=0,μ(12)=0.5,μ(34)=0.5,μ(13)=0.2,μ(24)=0.8,μ(Ω)=1. The probability μ on $D_{4}$ meets the extendability condition.*


Extendability proves to be more than a helpful mathematical property for embedding pre-Dynkin systems and their respective probabilities into algebras. Whether a probability defined on D can be extended to a probability on $2^{\Omega }$ is directly connected to the question whether the probability measure on D is coherent in the sense of [2] (pp. 68, 84) or not. Coherence is a minimal consistency requirement for probabilistic descriptions which has been introduced in the fundamental work of De Finetti [50] and developed by Walley [2]. Shortly summarizing, an incoherent imprecise probability is tantamount to an irrational betting behavior, thus the name. Thus, extendability is, besides its mathematical convenience, a desirable property of probabilities in pre-Dynkin settings.

We adapt here the definition of coherence of previsions in [2] (Definition 2.5.1) to probabilities.

**Definition 5** (Coherent Probability)**.***Let $A\subseteq 2^{\Omega }$ be an arbitrary collection of subsets. A set function $\underline{\nu }:A\to \left[0,1\right]$ is a* coherent lower probability *if and only if*
$$\begin{array}{c} \underset{\omega \in \Omega }{\sup}\sum_{i=1}^{j}\left(\chi_{A_{i}}\left(\omega \right)-\underline{\nu }\left(A_{i}\right)\right)-m\left(\chi_{A_{0}}\left(\omega \right)-\underline{\nu }\left(A_{0}\right)\right)\geq 0, \end{array}$$*for any non-negative n,m∈N and any $A_{0},A_{1},\ldots A_{n}\in A$. If A is closed under complement, the conjugate* coherent upper probability *is given by $\bar{\nu }:=\left(A\right)1-\underline{\nu }\left(A^{c}\right)$ for all A∈A. If furthermore $\bar{\nu }\left(A\right)=\underline{\nu }\left(A\right)$ for all A∈A, we call $\nu :=\bar{\nu }$ a*coherent additive probability.

At first sight, the Horn-Tarski condition given in Theorem 3 and the coherence condition presented here already appear similar. This becomes even more apparent in Walley’s reformulation of coherence for additive probabilities [2] (Theorem 2.8.7). In the following, we show that this superficial similarity is indeed based on a rigorous link. Surprisingly, Walley did not mention Horn and Tarski’s work in his foundational book.

**Theorem** **4**(Extendability Equals Coherence). *Let D be a pre-Dynkin system on Ω. A finitely additive probability measure μ on D is extendable to $2^{\Omega }$ if and only if it is a coherent additive probability on D.*

**Proof.** If μ is a coherent additive probability on D, then the linear extension theorem [2] (Theorem 3.4.2) applies. Hence, a coherent additive probability $\nu :2^{\Omega }\to \left[0,1\right]$ exists, such that ${\nu |}_{D}=\mu$. In particular, ν is a finitely additive probability following Definition 3 on $2^{\Omega }$ [2] (Theorem 2.8.9).For the converse direction, we observe that if μ possess an extension following Definition 4, then such an extension is a finitely additive probability on $2^{\Omega }$ following Definition 3. Hence, Walley [2] (Theorem 2.8.9) guarantees that the extension is a coherent additive probability (Definition 5). Any restriction to a subdomain $D\subseteq 2^{\Omega }$ keeps the probability coherent and additive. □

The linear extension theorem in Walley [2] (Theorem 3.4.2) used here is a generalization of de Finetti’s fundamental theorem of probability [50] (Theorems 3.10.1 and 3.10.7). De Finetti’s theorem is furthermore interesting, as he explicitly states that a coherent additive probability defined on an arbitrary collection of sets can be extended in a precise way (so lower and upper probability coincide) to some sets. De Finetti does not characterize this collection. Our Theorem 2, however, gives an answer to this question: the collection forms a pre-Dynkin system. It is possible to recover the statement by an argumentation using credal sets. Roughly, credal sets of coherent lower probabilities consist of all dominating coherent additive probabilities. In the case here, extendability guarantees that the credal set of a probability defined on a pre-Dynkin system is non-empty. Furthermore, Walley [2] introduced another slightly weaker concept than coherence, called “avoiding sure loss” which is equivalent to non-empty credal sets [2] (Corollary 3.3.4). Coherence and avoiding sure loss are equivalent for probabilities defined on pre-Dynkin systems [51] (Theorem 4.12).

Theorem 4 provides a missing link between two strands of work: on the one hand, probabilities on pre-Dynkin systems and related weak set structures have been closely investigated in foundational quantum probability theory [4,19] and decision theory [5,6]. On the other hand, coherent probabilities are central to imprecise probability, in particular, the more general formulations of coherent previsions and risk measures [2,52,53].

The reader familiar with the literature on imprecise probability might well not be surprised by the equivalence of extendability and coherence. We still think that this link is indeed valuable to be spelled out explicitly here. The concept of extendability and coherence have been developed separately in two communities with different goals in focus. Coherence tries to capture “rational” betting behavior [2,50]. Extendability links to what is sometimes called “quantum weirdness”. For readers familiar with quantum probability, we sketch the relationship between extendability and the concepts of compatibility and contextuality in the following.

#### 4.2.1. Extendability, Compatibility and Contextuality

Extendability in quantum theory tightly interacts with a series of properties and concepts which pervade discussions about the “specialness” of quantum theory in comparison to other classical physical theories: compatibility, contextuality, hidden variables and more. To be concrete, two measurements are compatible if, for any initial state, usually represented as a probability distribution (cf. [4]), there exists a joint measurement such that a fixed joint distribution for both measurement outcomes exists, whose marginals are the distribution of the single measurement [54,55]. We remark that this notion of compatibility is related, but not directly, to our Definition 2. If measurements are incompatible, then there are potentially still states such that a joint distribution of measurements exists. Only in the cases that no joint distribution of measurements exists, i.e., extendability is not provided, a measuring observer observes *contextual* behavior [55]. Translated to the language of imprecise probability, contextuality amounts to non-coherence of a probabilistic description. Compatibility, in contrast, is a structural notion. If any finitely additive probability on a pre-Dynkin system is extendable, then the pre-Dynkin system, very roughly, resembles compatible measurements. We are indeed not the first to notice intriguing links between imprecise probability and concepts therein to quantum mechanics. Benavoli and collaborators recovered the four postulates of quantum mechanics with desirability as a starting point [56] (Desirability is a relatively general framework for imprecise probability [57].)

#### 4.2.2. Extendability and Marginal Problem

A particularly helpful special case of probabilities defined on pre-Dynkin systems are marginal scenarios. Marginal scenarios, as we already stated in Section 2, are settings in which several (classical) marginal probability distributions, for instance, for three random variables $p_{X},p_{Y},p_{Z}$, are given, but a joint distribution $p_{X,Y,Z}$ is not specified. We note that we do not define random variables, marginals and probability distributions rigorously here. Instead, we argue on a vague level to convey the intuition. Such marginal scenarios can be expressed via probabilities on pre-Dynkin systems as laid out in detail in [34,58] and [33] (Example 4.2). Central to marginal scenarios is the relation structure of the marginals. For instance, $p_{X,Y},p_{Z}$ as well as $p_{X,Y},p_{Y,Z}$ or $p_{X,Y},p_{Y,Z},p_{X,Z}$ form marginal scenarios. The so-called marginal problem for marginal scenarios now asks whether for all instantiations of the marginal scenario there exists a joint distribution, e.g., for all $p_{X,Y},p_{Z}$ a joint distribution $p_{X,Y,Z}$ exists. In contrast, for some assignments $p_{X,Y},p_{Y,Z},p_{X,Z}$ there does not exist a corresponding $p_{X,Y,Z}$. (The attentive reader might have noticed the similarity to the notion of compatibility, for good reasons [59] (§V.B.2). Marginal scenarios are used to represent multi-measurement settings and compatibility among the measurements.) This question was, some while ago, asked for probabilities on finite spaces [34], countably additive probabilities [58], finitely additive probabilities (cf. [60]) and recently for even more general probability models—sets of desirable gambles [61,62]. A recurring theme in all those studies is the so-called running intersection property which characterizes all solvable marginal problems. In other words, the running intersection property guarantees that *every* instantiation of the marginal distributions is *extendable*. But there exist marginal problems for which only *specific* instantiations of the marginal distributions allow for extendability.

### 4.3. Coherent Extension

A probability on a pre-Dynkin system D, even when extendable, only allows for probabilistic statements on D itself. However, extendability guarantees that a “nice” embedding into a larger system of measurable sets exists. More specifically, extendability expressed in terms of credal sets provides a well-known tool for the worst-case extension of a probability from a pre-Dynkin system to a larger algebra.

If a finitely additive probability on a pre-Dynkin system is extendable, then we can obtain lower and upper probabilities of events which are not in the pre-Dynkin system but on a larger algebra. We follow the idea of natural extensions, e.g., as described by [2] (p. 136). In particular, [2] (Theorem 3.3.4 (b)) directly applies as long as a probability on a pre-Dynkin system is extendable.

**Corollary 1** (Coherent Extension of Probability)**.***Let D be a pre-Dynkin system on Ω. For a finitely additive probability measure μ on D we define the* credal set
$$\begin{array}{c} M\left(\mu ,D\right):=\left\{\nu \in \Delta :\nu (A)=\mu (A),\;\forall A\in D\right\}. \end{array}$$*If μ on D is extendable to $2^{\Omega }$, then $\forall A\in 2^{\Omega }$,*
$$\begin{array}{c} {\underline{\mu }}_{D}\left(A\right):=\underset{\nu \in M(\mu ,D)}{\inf}\nu \left(A\right),\;{\bar{\mu }}_{D}\left(A\right):=\underset{\nu \in M(\mu ,D)}{\sup}\nu \left(A\right). \end{array}$$*define a coherent lower respectively upper probability on $2^{\Omega }$.*

**Example** **8.**
*The coherent extension of μ on $D_{4}$ as defined in Example 7 is ${\underline{\mu }}_{D_{4}}=\ell$ and ${\bar{\mu }}_{D_{4}}=u$ where, ℓ and u are defined as in Example 4 (cf. [2] (p. 122)). Figure 2 illustrates the coherent extensions. Even though coherent, ${\underline{\mu }}_{D_{4}}$ is neither supermodular nor submodular:*

$$\begin{array}{c} {\underline{\mu }}_{D_{4}}\left(12\right)+{\underline{\mu }}_{D_{4}}\left(13\right)=0.5+0.2>0.5+0.0={\underline{\mu }}_{D_{4}}\left(123\right)+{\underline{\mu }}_{D_{4}}\left(1\right),\\ {\underline{\mu }}_{D_{4}}\left(1\right)+{\underline{\mu }}_{D_{4}}\left(2\right)=0.0+0.3<0.5+0.0={\underline{\mu }}_{D_{4}}\left(12\right)+{\underline{\mu }}_{D_{4}}\left(\emptyset \right). \end{array}$$

*This implies that as well ${\bar{\mu }}_{D_{4}}$ is neither supermodular nor submodular [43] (Proposition 2.3).*


These lower and upper probabilities allow for at least two interpretations: We can assume that a precise probability on a pre-Dynkin systems $D\subseteq 2^{\Omega }$ just reveals its values on D, but is actually defined over $2^{\Omega }$. Then the lower and upper probability constitute lower and upper bounds of the precise “hidden probability” on $2^{\Omega }$, which is solely accessible on D. On the other hand, we can even reject the existence of such precise “hidden probability”. Then lower and upper probability *are* the inherently imprecise probability of an event in $2^{\Omega }$ but not in D. (As remarked by Walley [2] (p. 138), De Finetti [50] surprisingly only considered the first mentioned interpretation.)

The obtained lower and upper probabilities represent the imprecise interdependencies between all events of precise probabilities. We illustrate this statement: in the variety of updating methods in imprecise probability we pick the generalized Bayes’ rule [2] (§6.4) to exemplarily compute the conditional probability of two events for the coherent extension of a probability from a pre-Dynkin system. For A,B∈D such that μ(B)>0, the generalized Bayes’ rule gives [2] (Theorem 6.4.2): $$\begin{array}{cc} {\bar{\mu }}_{D}\left(A|B\right):= & \underset{\nu \in M(\mu ,D)}{\sup}\frac{\nu (A\cap B)}{\nu (B)}=\frac{\sup_{\nu \in M(\mu ,D)}\nu \left(A\cap B\right)}{\mu (B)}=\frac{{\bar{\mu }}_{D}\left(A\cap B\right)}{\mu (B)},\;\forall A,B\in D \end{array}$$We can easily rearrange the above as ${\bar{\mu }}_{D}\left(A\cap B\right)={\bar{\mu }}_{D}\left(A|B\right)\mu \left(B\right)$. In this case the imprecision of the probability of the intersected event is *purely* controlled by the conditional probability $\bar{\mu }\left(A|B\right)$ and *not* by the marginal, which is precise. So, the imprecision captured by the lower and upper probabilities locates solely in the interdependency of the events. We remark that Dempster’s rule gives the same conditional probability here [63].

### 4.4. Inner and Outer Extension Is More Pessimistic Than Coherent Extension

We have presented two extension methods for probabilities defined on pre-Dynkin systems. We relate the methods in the following. In the case of an extendable probability we can guarantee the following inequalities to hold.

**Theorem** **5**(Extension Theorem—Finitely Additive Case). *Let D be a pre-Dynkin system on Ω and μ a finitely additive probability on D which is extendable to $2^{\Omega }$. Then,*
$$\begin{array}{c} \mu_{*}\left(A\right)\leq {\underline{\mu }}_{D}\left(A\right)\leq {\bar{\mu }}_{D}\left(A\right)\leq \mu^{*}\left(A\right),\;\forall A\in 2^{\Omega }. \end{array}$$

**Proof.** Since $D\subseteq 2^{\Omega }$, we easily obtain
$$\begin{array}{cc} \mu_{*}\left(A\right) & =\sup\{\mu (B):A⊇B\in D\}\\ \; & =\sup\left\{{\underline{\mu }}_{D}\left(B\right):A⊇B\in D\right\}\\ \; & \leq \sup\left\{{\underline{\mu }}_{D}\left(B\right):A⊇B\in 2^{\Omega }\right\}\\ \; & ={\underline{\mu }}_{D}\left(A\right), \end{array}$$
for all $A\in 2^{\Omega }$. The other inequalities follow by the conjugacy of inner and outer measure and lower and upper coherent extension. □

In words, Theorem 5 states that the inner and outer extension is more “pessimistic” than the coherent extension. We use “pessimistic” in the sense of giving a looser bound for the probabilities assigned to elements not in the pre-Dynkin system D but in $2^{\Omega }$. We remark that Walley has shown that for a probability defined on a set algebra, inner and lower coherent extension, as well as outer and upper coherent extension, coincide [2] (Theorem 3.1.5). Walley even characterizes the sets for which a unique coherent extension exists [2] (Theorem 3.1.9). In Appendix D, we demonstrate analogous inequalities for countably additive probabilities on Dynkin systems.

## 5. The Credal Set and Its Relation to Pre-Dynkin System Structure

In the earlier parts of the paper, we derived pre-Dynkin systems as the system of precision for relatively general imprecise probabilities. Then, we showed that, under extendability conditions, a precise probability on a pre-Dynkin system gives rise to a coherent imprecise probability on an encompassing algebra. In other words, imprecise probabilities can be “mapped” to pre-Dynkin systems and vice versa. We concretize these mappings in the following. This manifestation then reveals structure in the interplay between the systems of precision, i.e., pre-Dynkin systems, and coherent imprecise probabilities. In particular, we argue that the order structure of pre-Dynkin systems can be mapped to the space of finitely additive probabilities. This provides a (lattice) duality for coherent imprecise probabilities with precise probabilities on pre-Dynkin systems. More concretely, the duality allows for the interpolation from imprecise probabilities which are precise on “all” events to imprecise probabilities which are precise only on the empty set and the entire set.

In the following discussion, we assume, in addition to the technicalities presented in Section 1, that a fixed finitely additive probability on $2^{\Omega }$, which we call ψ, is given. The finitely additive probability ψ with the algebra $2^{\Omega }$ and the base set Ω constitute our “base measure space” analogous to the choice of a base measure space in the theory of coherent risk measures [52]. The probability ψ can be interpreted as the pivot point around which we define increasingly imprecise coherent lower and upper probabilities. The choice of ψ particularly influences the sets of measure zero, which will play a major role in characterizing so-called bipolar-closed set systems. In comparison to the previous sections, we use ψ instead of μ as “reference measure” to emphasize the difference that μ was defined on a relatively arbitrary pre-Dynkin systems D on Ω, while ψ is defined and fixed on the algebra $2^{\Omega }$ on Ω.

### 5.1. Credal Set Function Maps from Pre-Dynkin Systems to Coherent Probabilities

Equipped with a reference measure ψ we define the credal set function. The name arises due to its close link to the credal set as defined in Corollary 1.

**Definition 6** (Credal Set Function)**.***Let Δ be the set of all finitely additive probabilities on $2^{\Omega }$. For a fixed, finitely additive probability ψ∈Δ, we call*$$\begin{array}{c} m_{\psi }:2^{2^{\Omega }}\to 2^{\Delta },\;m_{\psi }\left(A\right):=\left\{\nu \in \Delta :\nu \left(A\right)=\psi \left(A\right),\;\forall A\in A\right\}, \end{array}$$*the* credal set function.

**Example** **9.**
*Let $\Omega_{4}=\left\{1,2,3,4\right\}$ as in Example 1. With abuse of notation, we define the probability $\psi :2^{\Omega_{4}}\to \left[0,1\right]$ via its corresponding point on the simplex ψ∈Δ, $\psi_{1}=0.2,\psi_{2}=0.3,\psi_{3}=0.5,\psi_{4}=0$. It follows, e.g., $m_{\psi }\left(\left\{12,3\right\}\right)=\left\{\nu \in \Delta :\nu_{1}+\nu_{2}=0.5,\nu_{3}=0.5\right\}$. In the subsequent examples, we implicitly assume the here defined ψ.*


We stress that although not notated explicitly, the credal set function depends upon the choice of ψ. For a fixed ψ on $2^{\Omega }$, the credal set function $m_{\psi }$ maps a subset of the algebra $2^{\Omega }$ to the set of all finitely additive probabilities which coincide with ψ on this subset. It should be noticed that by definition of ψ, $m_{\psi }\left(A\right)\neq \emptyset$ for every non-empty $A\subseteq 2^{\Omega }$, because $\psi \in m_{\psi }\left(A\right)$ for every $A\subseteq 2^{\Omega }$.

This defined mapping now simplifies our discussion about how pre-Dynkin systems and imprecise probabilities correspond. For instance, one can easily see that the extreme case $A=2^{\Omega }$ corresponds to $m_{\psi }\left(A\right)=\left\{\psi \right\}$ and A=∅ to $m_{\psi }\left(A\right)=\Delta$. More generally, we observe the following two properties of the credal set function.

**Proposition 2** (Credal Set Function is Invariant to Pre-Dynkin Hull)**.**
*Let $m_{\psi }$ be the credal set function. For any $A\subseteq 2^{\Omega }$,*

$$\begin{array}{c} m_{\psi }\left(A\right)=m_{\psi }\left(D\left(A\right)\right). \end{array}$$



**Proof.** We need to show that
$$\begin{array}{c} \{\nu \in \Delta :\nu (A)=\psi (A),\;A\in A\}=\{\nu \in \Delta :\nu (A)=\psi (A),\;A\in D(A)\}. \end{array}$$The set inclusion of the right hand side in the left hand side is trivial. For the reverse direction, consider an element ν∈Δ such that ν(A)=ψ(A) for A∈A. Let
$$\begin{array}{c} H:=\{A\in D(A):\nu (A)=\psi (A)\}. \end{array}$$By Theorem 2 H is a pre-Dynkin system. Since A⊆H we know D(A)⊆H. Hence, ν(A)=μ(A) for A∈D(A). This gives the desired inclusion. We remark that for A=∅ the equality still holds, since D(∅)={∅,Ω}. □

**Example** **10.**
*The credal set $m_{\psi }\left(\left\{12,3\right\}\right)=\left\{\nu \in \Delta :\nu_{1}+\nu_{2}=0.5,\nu_{3}=0.5\right\}$ given in Example 9 nicely illustrates Proposition 2:*

$$\begin{array}{c} m_{\psi }\left(\left\{12,3\right\}\right)=\left\{\nu \in \Delta :\nu_{1}+\nu_{2}=0.5,\nu_{3}=0.5,\nu_{4}=0\right\}=m_{\psi }\left(\left\{\emptyset ,12,3,4,34,123,124,\Omega_{4}\right\}\right). \end{array}$$



**Proposition 3** (Credal Set Function Maps to Weak${}^{★}$-Closed Convex Sets)**.**
*Let $m_{\psi }$ be the credal set function. For every non-empty $A\subseteq 2^{\Omega }$, $m_{\psi }\left(A\right)$ is weak${}^{★}$-closed convex.*


**Proof.** The reference probability ψ is by definition coherent. Hence, for all non-empty $A\subseteq 2^{\Omega }$, the set $m_{\psi }\left(A\right)$ is the set of all ν∈Δ which dominate ψ. This set is, by Theorem 3.6.1 in [2], weak${}^{★}$-closed and convex. □

In words, the credal set on some set system coincides with the credal set on its generated pre-Dynkin system. And the credal set of probabilities is always weak${}^{★}$-closed and convex. Proposition 2 allows us to work with credal sets of arbitrary set systems instead of the entire pre-Dynkin system. Thus, it resembles the well-known π-λ-Theorem, which is fundamental to classical probability theory [23] (Lemma A.1.3). On the other hand, this result justifies our focus on pre-Dynkin systems instead of arbitrary set systems. We do not lose generality when considering pre-Dynkin systems instead of non-structured sets of sets.

Proposition 3 guarantees that the images of the credal set function behave “nicely”. Specifically, these weak${}^{★}$-closed convex sets correspond to coherent previsions, i.e., generalizations of coherent probabilities as already stated by Walley [2] (Theorem 3.6.1). We elaborate this observation in Section 5.4. In conclusion, credal set functions map pre-Dynkin systems to coherent probabilities. What about the reverse mapping?

### 5.2. The Dual Credal Set Function

The following is a natural definition of a dual credal set function. We justify this name by Proposition 4 below.

**Definition 7** (Dual Credal Set Function)**.***Let *Δ* be the set of all finitely additive probabilities on $2^{\Omega }$. Fix a finitely additive probability ψ on $2^{\Omega }$. We call*$$\begin{array}{c} m_{\psi }^{\circ }:2^{\Delta }\to 2^{2^{\Omega }},\;m_{\psi }^{\circ }\left(Q\right):=\left\{A\in 2^{\Omega }:\nu \left(A\right)=\psi \left(A\right),\forall \nu \in Q\right\} \end{array}$$*the* dual credal set function.

The dual credal set function also depends upon ψ, but we do not notate this explicitly. The dual credal set function maps an arbitrary set of finitely additive probability measures on $2^{\Omega }$ to the (largest) set of events on which all contained probabilities coincide. We remark that each set of finitely additive probability measures can be linked to an imprecise probability.

We suggestively called the antagonist to the credal set function the “dual credal set function”. The duality appearing here is a well-known fundamental relationship between partially ordered sets: a Galois connection. A *Galois connection* is a pair of mappings f:X→Y and $f^{\circ }:Y\to X$ on partially ordered sets (Y,≤) and (X,≤), which preserves order structure (cf. Corollary 2). More formally, *f* and $f^{\circ }$ are a Galois connection if and only if for all x∈X,y∈Y, $x\leq f\left(y\right)\Leftrightarrow y\leq f^{\circ }\left(x\right)$ [64] (§V.8). Galois connections, even though they do *not* form an order isomorphism, induce a lattice duality. We exploit this lattice duality to provide an order-theoretic interpolation from no compatibility at all to full compatability. For this purpose, we first establish the Galois connection.

**Proposition 4** (Galois Connection by (Dual) Credal Set Function)**.**
*The credal set function $m_{\psi }$ and the dual credal set function $m_{\psi }^{\circ }$ form a Galois connection.*


**Proof.** $m_{\psi }$ and $m_{\psi }^{\circ }$ form a Galois connection if and only if $A\subseteq m_{\psi }^{\circ }\left(Q\right)\Leftrightarrow Q\subseteq m_{\psi }\left(A\right)$ [64] (§V.8). First, we show the left to right implication. We assume $A\subseteq m_{\psi }^{\circ }\left(Q\right)$, i.e. every ν∈Q coincides with ψ on A. Hence,
$$\begin{array}{cc} \nu \in Q & \Rightarrow \nu (A)=\psi (A),\forall A\in A\\ \; & \Rightarrow \nu \in \left\{\nu^{\prime }\in \Delta :\nu^{\prime }\left(A\right)=\psi \left(A\right),\forall A\in A\right\}=m_{\psi }\left(A\right). \end{array}$$In case of the right to left implication, we suppose $Q\subseteq m_{\psi }\left(A\right)$. Thus,
$$\begin{array}{cc} A\in A & \Rightarrow \nu (A)=\psi (A),\forall \nu \in Q\\ \; & \Rightarrow A\in \left\{A^{\prime }\in 2^{\Omega }:\nu \left(A^{\prime }\right)=\psi \left(A^{\prime }\right),\forall \nu \in Q\right\}=m_{\psi }^{\circ }\left(Q\right). \end{array}$$□

The mappings involved in the Galois connection are antitone, i.e., they reverse the order structure from domain to codomain. Their pairwise application is extensive, i.e., the image of an object contains the object. In summary, the following rules of calculation hold:

**Corollary 2** (Rules for (Dual) Credal Set Function)**.**
*Let $m_{\psi }$ be the credal set function and $m_{\psi }^{\circ }$ be the dual credal set function. For arbitrary $A_{1},A_{2},A\subseteq 2^{\Omega }$ and $Q_{1},Q_{2},Q\subseteq \Delta$,*

$$\begin{array}{ccccc} A_{1}\subseteq A_{2} & \Rightarrow m_{\psi }\left(A_{2}\right)\subseteq m_{\psi }\left(A_{1}\right), & Q_{1}\subseteq Q_{2} & \Rightarrow m_{\psi }^{\circ }\left(Q_{2}\right)\subseteq m_{\psi }^{\circ }\left(Q_{1}\right), & \left(\mathrm{antitone}\right)\\ Q & \subseteq m_{\psi }\left(m_{\psi }^{\circ }\left(Q\right)\right), & A & \subseteq m_{\psi }^{\circ }\left(m_{\psi }\left(A\right)\right), & \left(\mathrm{extensive}\right)\\ m_{\psi }\left(A\right) & =m_{\psi }\left(m_{\psi }^{\circ }\left(m_{\psi }\left(A\right)\right)\right), & m_{\psi }^{\circ }\left(Q\right) & =m_{\psi }^{\circ }\left(m_{\psi }\left(m_{\psi }^{\circ }\left(Q\right)\right)\right), & (\mathrm{pseudo}-\mathrm{inverse}). \end{array}$$



**Proof.** [64] (§V.7 and V.8) □

Proposition 4 provides a tool to further investigate the dual credal set function. The reader might have noticed the similarity of the dual credal set function and the main question of Section 3: given a lower and upper probability, on which set systems do both coincide? In fact, we obtain an analogous result to Theorem 2, and again an imprecise probability is mapped to the set of events on which it is precise.

**Proposition 5** (Dual Credal Set Function Maps to Pre-Dynkin Systems)**.**
*Let $m_{\psi }^{\circ }$ be the dual credal set function. For all non-empty Q⊆Δ, $m_{\psi }^{\circ }\left(Q\right)$ is a pre-Dynkin system.*


**Proof.** We show the statement by establishing the equality $D\left(m_{\psi }^{\circ }\left(Q\right)\right)=m_{\psi }^{\circ }\left(Q\right)$. Trivially, $D\left(m_{\psi }^{\circ }\left(Q\right)\right)⊇m_{\psi }^{\circ }\left(Q\right)$. Furthermore,
$$\begin{array}{cc} D(m_{\psi }^{\circ }\left(Q\right)) & \overset{C.2}{\subseteq }m_{\psi }^{\circ }\left(m_{\psi }\left(D\left(m_{\psi }^{\circ }\left(Q\right)\right)\right)\right)\\ \; & \overset{P.2}{=}m_{\psi }^{\circ }\left(m_{\psi }\left(m_{\psi }^{\circ }\left(Q\right)\right)\right)\overset{C.2}{=}m_{\psi }^{\circ }\left(Q\right). \end{array}$$□

**Proposition 6** (Dual Credal Set Function is Invariant to Weak${}^{★}$-Closed Convex Hull)**.**
*Let $m_{\psi }^{\circ }$ be the dual credal set function. For any Q⊆Δ, $m_{\psi }^{\circ }\left(Q\right)=m_{\psi }^{\circ }\left(\bar{\mathrm{co}}\;Q\right)$.*


**Proof.** By Corollary 2, $m_{\psi }^{\circ }\left(Q\right)=m_{\psi }^{\circ }\left(m_{\psi }\left(m_{\psi }^{\circ }\left(Q\right)\right)\right)$. Furthermore, $Q\subseteq m_{\psi }\left(m_{\psi }^{\circ }\left(Q\right)\right)$. Via, Proposition 3 we obtain $\bar{\mathrm{co}}\;Q\subseteq m_{\psi }\left(m_{\psi }^{\circ }\left(Q\right)\right)$. Hence, the result follows. □

**Example** **11.**
*Let Q={ν} where we identify the probability ν on $2^{\Omega_{4}}$ with an element $\nu \in \Delta_{4}$. For instance, $\nu_{1}=0,\nu_{2}=0.5,\nu_{3}=0,\nu_{4}=0.5$. It is easy to see that $m_{\psi }^{\circ }\left(Q\right)=D\left(\left\{12,3\right\}\right)$ as given in Example 10.*


In summary, credal set functions map pre-Dynkin systems to weak${}^{★}$-closed convex credal sets. Dual credal set functions map (weak${}^{★}$-closed convex) credal sets to pre-Dynkin systems. In addition, the two functions form a Galois connection. In fact, every Galois connection defines closure operators, i.e., extensive, monotone and idempotent maps [25] (Definition 4.5.a). The closure operators are defined as the sequential application of the credal set function and the dual credal set function to subsets of Δ or $2^{\Omega }$. In symbols: $A\mapsto m_{\psi }^{\circ }\left(m_{\psi }\left(A\right)\right),Q\mapsto m_{\psi }\left(m_{\psi }^{\circ }\left(Q\right)\right)$. In particular, these closure operators define bipolar-closed sets.

### 5.3. Bipolar-Closed Sets

Bipolar-closed sets are sets $A\subseteq 2^{\Omega }$ such that $A=m_{\psi }^{\circ }\left(m_{\psi }\left(A\right)\right)$, respectively, Q⊆Δ such that $Q=m_{\psi }\left(m_{\psi }^{\circ }\left(Q\right)\right)$. Most importantly, the bipolar-closed sets form two antitone isomorphic lattices ordered by set inclusion [64] (Theorem V.8.20). This relationship gives us a lattice duality between set systems and credal sets of probabilities. See Figure 3 for an illustration of bipolar-closed sets and the Galois connection.

**Example** **12.**
*The pre-Dynkin system D({12,3}) already discussed in Example 10 and Example 11 is a bipolar-closed set. In contrast, the set {12,3} cannot be a bipolar-closed set, as it is not a pre-Dynkin system.*


More precisely, bipolar-closed sets in the set of finitely additive probability distributions are weak${}^{★}$-closed convex (Proposition 3). These map to bipolar-closed subsets of $2^{\Omega }$, which are pre-Dynkin systems (Proposition 5). All of the stated properties of bipolar-closed sets are necessary. But are they sufficient?

#### Sufficient Conditions for Bipolar-Closed Sets

In the search for sufficient conditions for bipolar-closed sets we focus on bipolar-closed subsets of $2^{\Omega }$. We leave bipolar-closed subsets of Δ to future investigations. Already, bipolar-closed subsets of $2^{\Omega }$ are surprisingly difficult to characterize. We show that bipolar-closed sets are exactly the pre-Dynkin systems if the reference probability measure has support on all elements of a finite set. To this end, let us start with a simple implication of Proposition 2 and Corollary 2.

**Corollary** **3.**
*Let $m_{\psi }$ be the credal set function and $m_{\psi }^{\circ }$ be the dual credal set function. For an arbitrary subset $A\subseteq 2^{\Omega }$ we have*

$$\begin{array}{c} D\left(A\right)\subseteq m_{\psi }^{\circ }\left(m_{\psi }\left(A\right)\right). \end{array}$$



**Proof.** By Proposition 2, $m_{\psi }^{\circ }\left(m_{\psi }\left(A\right)\right)=m_{\psi }^{\circ }\left(m_{\psi }\left(D\left(A\right)\right)\right)$. By Corollary 2, the statement follows. □

This corollary gives rise to the follow-up question: under which circumstances does $D\left(A\right)=m_{\psi }^{\circ }\left(m_{\psi }\left(A\right)\right)$? As the following theorem demonstrates, this question is closely connected to the sets of measure zero of the base probability ψ and its problems (cf. [65]).

**Proposition 7** (“Closedness” under Measure Zero Sets)**.**
*Let $m_{\psi }$ be the credal set function and $m_{\psi }^{\circ }$ be the dual credal set function. Let $A\subseteq 2^{\Omega }$. For any $A\in m_{\psi }^{\circ }\left(m_{\psi }\left(A\right)\right)$, if ψ(A)=0, then all $B,C\in 2^{\Omega }$ such that B⊆A and $C⊇A^{c}$ are in $m_{\psi }^{\circ }\left(m_{\psi }\left(A\right)\right)$.*


**Proof.** Let $A\in m_{\psi }^{\circ }\left(m_{\psi }\left(A\right)\right)$ for arbitrary $A\subseteq 2^{\Omega }$ with ψ(A)=0. Consider two sets $B,C\in 2^{\Omega }$ such that B⊆A, respectively, $C⊇A^{c}$. Then, ψ(B)=0 and ψ(C)=1. Furthermore, for any $\nu \in m_{\psi }\left(m_{\psi }^{\circ }\left(m_{\psi }\left(A\right)\right)\right)$, we have ν(B)=0 (respectively, ν(C)=1). Since $m_{\psi }\left(m_{\psi }^{\circ }\left(m_{\psi }\left(A\right)\right)\right)=m_{\psi }\left(A\right)$, we obtain $B,C\in m_{\psi }^{\circ }\left(m_{\psi }\left(A\right)\right)$. □

A pre-Dynkin system D(A) can only coincide with $m_{\psi }^{\circ }\left(m_{\psi }\left(A\right)\right)$ if subsets of measure zero are included. Thus, Proposition 7 provides a further necessary condition for bipolar-closed subsets of $2^{\Omega }$. Yet, it turns out that the sets of measure zero as well can give a sufficient condition for bipolar-closed sets at least in a finite setting.

**Theorem** **6**(Pre-Dynkin Hull is Bipolar-Closure in Finite, Discrete Setting). *Let Ω=[n]. Fix a finitely additive probability ψ on $2^{\Omega }$ such that ψ(A)>0 for every $A\in 2^{\Omega }\setminus \left\{\emptyset \right\}$. Let $D\subseteq 2^{\Omega }$ be a pre-Dynkin system. Then,*
$$\begin{array}{c} D=m_{\psi }^{\circ }\left(m_{\psi }\left(D\right)\right). \end{array}$$

**Proof.** If $D=2^{\left[n\right]}$ the result follows directly. Thus, from now on we assume $D⊊2^{\left[n\right]}$. The set inclusion $D\subseteq m_{\psi }^{\circ }\left(m_{\psi }\left(D\right)\right)$ is given by Corollary 3. We show $D⊇m_{\psi }^{\circ }\left(m_{\psi }\left(D\right)\right)$ by proving that for every $B\in 2^{\left[n\right]}\setminus D$ there exists $\nu \in m_{\psi }\left(D\right)$ such that ν(B)≠ψ(B).Let us consider an arbitrary $B\in 2^{\left[n\right]}\setminus D$. Without loss of generality (Lemma A2) we can decompose
$$\begin{array}{c} B=B_{D}\cup A, \end{array}$$
where $B_{D}\in D$ and A∉D is a weak atom with respect to D (Definition A3). Thus, we can leverage Lemma A3: there exists $\nu \in m_{\psi }\left(D\right)$ such that ν(A)≠ψ(A). Thus,
$$\begin{array}{c} \nu \left(B\right)=\nu \left(B_{D}\right)+\nu \left(A\right)=\psi \left(B_{D}\right)+\nu \left(A\right)\neq \psi \left(B_{D}\right)+\psi \left(A\right). \end{array}$$It follows that $B\notin m_{\psi }^{\circ }\left(m_{\psi }\left(D\right)\right)$, concluding the proof. □

We emphasize that the statement does not hold as long as there are non-empty subsets of Ω which get assigned zero probability.

**Example 13** (Bipolar-Closure is Strictly Greater Than Pre-Dynkin Hull)**.**
*Let $\Omega_{3}=\left\{1,2,3\right\}$ and ψ({1})=1,ψ({2})=0,ψ({3})=0 constitute a base probability space. We have $D\left(\left\{1\right\}\right)=\left\{\emptyset ,1,23,\Omega_{3}\right\}$, but $m_{\psi }^{\circ }\left(m_{\psi }\left(\left\{1\right\}\right)\right)=2^{\Omega }$.*


Whether this theorem can be extended to more general sets Ω is an open question. Seemingly, proofs along the line of Theorem 6 are doomed to fail, since one cannot argue via probabilities on atoms of Ω.

### 5.4. Interpolation from Algebra to Trivial Pre-Dynkin System

In probability theory there is a choice to be made regarding which events should get assigned probabilities [1] (p. 52). This significant choice has (mathematically) been standardized to form a (σ-)algebra (cf. standard probability space). But, already Kolmogorov, the “father” of modern probability theory, emphasized that this choice is not universal but should depend on the problem at hand. More recently, Khrennikov [66] argued that a more appropriate probabilistic modeling should appeal to weaker domains for probabilities to, for instance, represent physical observations such as quantum phenomena.

In particular, it cannot always be taken for granted that all events are compatible with all others, as implied by a (σ-)algebra (cf. Section 2.1). For instance, von Mises’ axiomatization of probability inherently reflects potential incompatible events in terms of a pre-Dynkin system [20,29]. In other words, there is a choice to be made about the system of precision. Which sets should be compatible to each other, and which should not? How do the choices of the systems of precision relate to each other?

We neglect, without loss of generality, arbitrary systems of precision and focus on pre-Dynkin systems (cf. Proposition 2). The range of choices is captured by the system of pre-Dynkin systems.

**Proposition 8** (Set of Pre-Dynkin Systems is a Lattice)**.**
*The set $D:=\{D\subseteq 2^{\Omega }:D=D\left(D\right)\}$ is ordered by set inclusion. Furthermore, (D,⊆) is a lattice with*

$$\begin{array}{cc} \overset{n}{\underset{i=1}{⋁}}D_{i} & =D\left(\overset{n}{\underset{i=1}{⋃}}D_{i}\right)\\ \overset{n}{\underset{i=1}{⋀}}D_{i} & =\overset{n}{\underset{i=1}{⋂}}D_{i}. \end{array}$$



**Proof.** On the one hand, it is easy to show that the intersection of pre-Dynkin systems forms a pre-Dynkin system again. On other hand, the smallest pre-Dynkin system which contains a finite set of pre-Dynkin systems is by definition the pre-Dynkin system generated by the union over all elements in this finite set of pre-Dynkin systems. □

**Example** **14.**
*Let $\Omega_{4}$ be as defined in Example 9. The minimal element in D then is $\left\{\emptyset ,\Omega_{4}\right\}$. The maximal element is $2^{\Omega_{4}}$. For the sake of brevity, we omit all further elements in D and remain with the observation that $D_{4}$ of Example 1 and D({12,3}) are elements of D.*


The lattice D spans a range of choices from $D=2^{\Omega }$, i.e., complete compatibility and only a single probability distribution in its credal set, namely $m_{\psi }\left(2^{\Omega }\right)=\left\{\psi \right\}$, to D={∅,Ω}, i.e., no compatibility and the entire space of probability distributions constitute its credal set $m_{\psi }\left(\left\{\emptyset ,\Omega \right\}\right)=\Delta$. How “close” D is to the algebra $2^{\Omega }$ determines how “classical” the credal set behaves. In other words, (D,⊆) parametrizes a family of credal sets. Thus, it parametrizes coherent probabilities. The knob of compatibility can be turned from trivially nothing ({∅,Ω}), to everything ($2^{\Omega }$). How does the “amount of compatibility” of the pre-Dynkin system map to the credal sets? Or, e.g., given two pre-Dynkin systems on which a probability is defined, what is the credal set of the union of these systems?

**Proposition 9** (Lattice of Dynkin Systems and Credal Sets)**.**
*The credal set function $m_{\psi }$ (Definition 6) together with the lattice (D,⊆) provides a parametrized family of credal sets for which hold ($\forall D_{1},D_{2}\in D$):*

$$\begin{array}{cc} m_{\psi }\left(D_{1}\vee D_{2}\right) & =m_{\psi }\left(D_{1}\cup D_{2}\right)=m_{\psi }\left(D_{1}\right)\cap m_{\psi }\left(D_{2}\right)\\ m_{\psi }\left(D_{1}\wedge D_{2}\right) & =m_{\psi }\left(D_{1}\cap D_{2}\right)⊇m_{\psi }\left(D_{1}\right)\cup m_{\psi }\left(D_{2}\right). \end{array}$$



**Proof.** Concerning the first equality, we observe that for arbitrary $D_{1},D_{2}\in D$
$$\begin{array}{c} m_{\psi }\left(D_{1}\vee D_{2}\right)=m_{\psi }\left(D\left(D_{1}\vee D_{2}\right)\right)=m_{\psi }\left(D_{1}\cup D_{2}\right), \end{array}$$
by Proposition 2. Consequently,
$$\begin{array}{cc} m_{\psi }\left(D_{1}\cup D_{2}\right) & =\{\nu \in \Delta :\nu \left(D\right)=\psi \left(D\right),\;\forall D\in D_{1}\cup D_{2}\}\\ \; & =\left\{\nu \in \Delta :\nu \left(D\right)=\psi \left(D\right),\;\forall D\in D_{1}\right\}\cap \left\{\nu \in \Delta :\nu \left(D\right)=\psi \left(D\right),\;\forall D\in D_{2}\right\}\\ \; & =m_{\psi }\left(D_{1}\right)\cap m_{\psi }\left(D_{2}\right). \end{array}$$The second line follows by the definition of infimum on the lattice of pre-Dynkin systems and simple set containment: $m_{\psi }\left(D_{1}\right)\subseteq m_{\psi }\left(D_{1}\cap D_{2}\right)$ and $m_{\psi }\left(D_{2}\right)\subseteq m_{\psi }\left(D_{1}\cap D_{2}\right)$. □

Unfortunately, the mentioned interpolation is slightly improper. It turns out that there are pre-Dynkin systems $D_{1}\neq D_{2}$ such that $m_{\psi }\left(D_{1}\right)=m_{\psi }\left(D_{2}\right)$.

**Example 15** (Non-Injectivity of Credal Set Function)**.**
*Let $\Omega_{3}=\left\{1,2,3\right\}$ and ψ({1})=1,ψ({2})=0,ψ({3})=0 constitute a base probability space, cf. Example 13. Then, obviously $m_{\psi }\left(D\left(\left\{1\right\}\right)\right)=m_{\psi }\left(2^{\Omega }\right)$, but $D\left(\left\{1\right\}\right)\neq 2^{\Omega }$.*


The reason for this collision of credal sets is that not every pre-Dynkin system $D\subseteq 2^{\Omega }$ is a bipolar-closed set.

**Proposition 10** (Credal Set Function is Injective on Bipolar-Closed Sets)**.**
*Let $m_{\psi }$ be the credal set and $m_{\psi }^{\circ }$ be the dual credal set function. Let $D_{1},D_{2}\in D$ be pre-Dynkin systems, which are bipolar-closed. If $D_{1}\neq D_{2}$, then $m_{\psi }\left(D_{1}\right)\neq m_{\psi }\left(D_{2}\right)$.*


**Proof.** We prove the claim by contraposition:
$$\begin{array}{c} m_{\psi }\left(D_{1}\right)=m_{\psi }\left(D_{2}\right)\Rightarrow m_{\psi }^{\circ }\left(m_{\psi }\left(D_{1}\right)\right)=m_{\psi }^{\circ }\left(m_{\psi }\left(D_{2}\right)\right)\Rightarrow D_{1}=D_{2}. \end{array}$$□

Hence, it is reasonable to focus on the set of bipolar-closed sets contained in $2^{\Omega }$. We define the *set of interpolating pre-Dynkin systems* $C:=\{C\subseteq 2^{\Omega }:C=m_{\psi }^{\circ }\left(m_{\psi }\left(C\right)\right)\}$. We know that C⊆D (Proposition 5) and C is even a lattice contained in D.

**Proposition 11** (Lattice of Bipolar-Closed Sets and Credal Sets)**.**
*Let $m_{\psi }$ be the credal set function and $m_{\psi }^{\circ }$ be the dual credal set function. The set of interpolating pre-Dynkin systems C equipped with the *⊆*-ordering forms a lattice:*

$$\begin{array}{cc} \overset{n}{\underset{i=1}{⋁}}C_{i} & =m_{\psi }^{\circ }\left(m_{\psi }\left(\overset{n}{\underset{i=1}{⋃}}C_{i}\right)\right)\\ \overset{n}{\underset{i=1}{⋀}}C_{i} & =\overset{n}{\underset{i=1}{⋂}}C_{i}. \end{array}$$

*In particular, this lattice C is antitone isomorphic to the lattice of bipolar-closed sets in $2^{\Delta }$. It holds (We denote the composition of functions with *∘*.):*

$$\begin{array}{cc} m_{\psi }\left(C_{1}\vee C_{2}\right) & =m_{\psi }\left(C_{1}\cup C_{2}\right)=m_{\psi }\left(C_{1}\right)\cap m_{\psi }\left(C_{2}\right)\\ m_{\psi }\left(C_{1}\wedge C_{2}\right) & =m_{\psi }\left(C_{1}\cap C_{2}\right)=\left(m_{\psi }\circ m_{\psi }^{\circ }\right)\left(m_{\psi }\left(C_{1}\right)\cup m_{\psi }\left(C_{2}\right)\right). \end{array}$$



**Proof.** By Theorem V.8.20 in [64], C is a lattice and $m_{\psi }$ an antitone lattice isomorphism on C. The equations hold by simple manipulations
$$\begin{array}{c} m_{\psi }\left(C_{1}\vee C_{2}\right)=m_{\psi }\left(m_{\psi }^{\circ }\left(m_{\psi }\left(C_{1}\cup C_{2}\right)\right)\right)=m_{\psi }\left(C_{1}\cup C_{2}\right)=m_{\psi }\left(C_{1}\right)\cap m_{\psi }\left(C_{2}\right), \end{array}$$
and
$$\begin{array}{c} m_{\psi }\left(C_{1}\wedge C_{2}\right)=m_{\psi }\left(m_{\psi }^{\circ }\left(m_{\psi }\left(C_{1}\cap C_{2}\right)\right)\right)=\left(m_{\psi }\circ m_{\psi }^{\circ }\right)\left(m_{\psi }\left(C_{1}\right)\cup m_{\psi }\left(C_{2}\right)\right). \end{array}$$□

Note that C is not generally a sublattice of D. It is a lattice contained in the lattice D, but the closure operator for the supremum is distinct. Interestingly, the order structure which both lattices, D and C, induce on the set of credal sets via $m_{\psi }$ is identical. Every set $m_{\psi }\left(A\right)$ for arbitrary $A\subseteq 2^{\Omega }$ is bipolar-closed (Corollary 2). Hence, the lattice of bipolar-closed sets in $2^{\Delta }$ is the domain of $m_{\psi }$ for elements in D and C. In other words, the lattices D and C provide one and the same parametrized family of credal sets, thus one and the same parametrized family of imprecise probabilities.

In comparison to other parametrized families of imprecise probability, such as distortion risk measures [67], which heavily rely on convex analysis, the duality used here is structurally weaker. Lattice isomorphisms give a glimpse of structure to the involved dual spaces. Convex dualities as exploited in [68] are far more informative, but apparently not able to handle the structural knob which we presented in this work: the set of sets which get assigned precise probabilities. Nevertheless, a natural question arises from this lattice duality: how does this lattice duality relate to a convex duality? We leave this question open to further research. A first attempt to an answer is discussed in Appendix C, where we link the parametrized family of distorted probabilities to the pre-Dynkin system family of imprecise probabilities.

## 6. A More General Perspective—The Set of Gambles with Precise Expectation

To this point, we have exclusively focused on probabilities and set systems of events to which we assign probabilities. In fact, there is a more general story to be told. In the literature on imprecise probability, focus often lies on expectation-type functionals instead of probabilities and on sets of gambles (bounded functions from the base set Ω to the real numbers) instead of sets of events. One can easily see that the latter is more general and can recover the former. Indicator functions of events are gambles. An expectation-type functional evaluated on an indicator gamble of an event corresponds to a generalized probability of the event. The converse direction, i.e., recovering a unique expectation-type functional from an imprecise probability, however, is not always possible [2] (§2.7.3). In the following, we reiterate several questions which we asked in the preceding sections for probabilities and set systems.

### 6.1. Partial Expectations Generalize Finitely Additive Probabilities on (Pre-)Dynkin Systems

We propose the following definition of partial expectation and show afterwards that it is a natural generalization of finitely additive probabilities defined on (pre-)Dynkin systems.

**Definition 8** (Partial Expectation)**.***Let ${\left\{L_{i}\right\}}_{i\in I}$ be a non-empty family of linear subspaces of B(Ω). We call $E:{⋃}_{i\in I}L_{i}\to R$ a* partial expectation *if and only if all of the following conditions are fulfilled:*
*(a)* 
*for any i∈I and for all $f,g\in L_{i}$, then E(f+g)=E(f)+E(g), (Partial Linearity),*
*(b)* 
*for any i∈I and any $f\in L_{i}$, then E(f)≥inff, (Coherence).*



We remark that for this definition, we leveraged the requirements for a linear prevision (Definition 9) on a linear space given in [2] (Theorem 2.8.4). In other words, a partial expectation is a functional which is defined on a union of linear subspaces and behaves like a “classical” (finitely additive) expectation on each of the subspaces but not necessarily on all simultaneously. It is a linear prevision when restricted to one of the subspaces $L_{i}$ (cf. Definition 9).

There is a one-to-one correspondence of linear previsions and coherent additive probabilities. For every coherent additive probability ν defined on an algebra A, there is a unique linear prevision, which we equivalently denote ν, the set of all A-measurable gambles, which agrees with the probability ν on the indicator gambles of the sets in A [2] (Theorem 3.2.2). The following Proposition exploits this correspondence. A finitely additive probability defined on a pre-Dynkin system relates one-to-one to a partial expectation which is defined on the set of linear spaces induced by the simple gambles on the blocks of the pre-Dynkin system. To this end, we introduce the following two notations: let $F\subseteq 2^{\Omega }$ be an algebra. Then, S(Ω,F)⊆B(Ω) denotes the linear subspace of simple gambles on F, i.e., scaled and added indicator gambles of a finite number of disjoint sets (cf. [26] (Definition 4.2.12)). Let $F_{\sigma }\subseteq 2^{\Omega }$ be a σ-algebra. Then $B\left(\Omega ,F_{\sigma }\right)\subseteq B\left(\Omega \right)$ denotes the linear subspace of all bounded, real-valued, $F_{\sigma }$-measurable gambles.

**Proposition 12** (Finitely Additive Probability on Pre-Dynkin System and its Partial Expectation)**.**
*Let $D\subseteq 2^{\Omega }$ be a pre-Dynkin system. Let μ:D→[0,1] be a finitely additive probability defined on the pre-Dynkin system with block structure ${\left\{A_{i}\right\}}_{i\in I}$. Then, μ is in one-to-one correspondence to a partial expectation $E:{⋃}_{i\in I}S\left(\Omega ,A_{i}\right)\to R$ defined on the union of linear spaces of simple gambles induced by all blocks $A_{i}$ of D.*


**Proof.** By Theorem 1, we can decompose the pre-Dynkin system D into a set of blocks ${\left\{A_{i}\right\}}_{i\in I}$. Since, $A_{i}\subseteq 2^{\Omega }$ for all i∈I, each of the blocks induces a the linear subspace of simple gambles $S\left(\Omega ,A_{i}\right)\subseteq B\left(\Omega \right)$. Given a finitely additive measure μ:D→[0,1], we now define $E:{⋃}_{i\in I}S\left(\Omega ,A_{i}\right)\to R$ by
$$\begin{array}{c} {E\left(f\right):=\int fd\mu |}_{A_{i}}\;\mathrm{if}\;f\in S\left(\Omega ,A_{i}\right). \end{array}$$For every i∈I, ${E|}_{S(\Omega ,A_{i})}$ is a linear prevision in one-to-one correspondence to the finitely additive probability ${\mu |}_{A_{i}}$ [2] (Theorem 3.2.2). Hence, conditions (a) and (b) in Definition 8 are met [2] (Theorem 2.8.4). For $f\in S\left(\Omega ,A_{i}\right)\cap S\left(\Omega ,A_{j}\right)$ (i≠j,i,j∈I), we know that $f\in S(\Omega ,A_{i}\cap A_{j})$ (Lemma A4), hence,
$$\begin{array}{c} {\int fd\mu |}_{A_{i}}=\int fd\mu {{|}_{A_{i}\cap A_{j}}=\int fd\mu |}_{A_{j}}. \end{array}$$Thus, *E* is well-defined and there is no other partial expectation which agrees with μ on the indicator gambles of the sets in D. □

The attentive reader might have noticed that we defined the partial expectation in Proposition 12 on very specific linear subspaces of B(Ω), namely the linear subspaces of simple gambles. In fact, the statement would still hold when enlarging the linear subspaces of simple gambles $S(\Omega ,A_{i})$ for every i∈I to linear subspaces of functions which are “convergence in measure”-approximated by gambles in $S(\Omega ,A_{i})$. For more details, we refer the reader to [26] (Definition 4.4.5 and Corollary 4.4.9).

But, it is not the case that we can extend the definition to all sets of bounded, $A_{i}$-measurable functions, i.e., bounded functions whose pre-images of sets in the smallest algebra which contains all open sets of the real numbers are contained in $A_{i}$. For an algebra $A\subseteq 2^{\Omega }$, the set of A-measurable gambles is not necessarily a linear subspace of B(Ω) [2] (p. 129).

This is different for a σ-algebra $A_{\sigma }\subseteq 2^{\Omega }$. The set of bounded, $A_{\sigma }$-measurable functions $B(\Omega ,A_{\sigma })$ forms a linear subspace of B(Ω)[2] (p. 129). Here, measurability is defined as the pre-image of every Borel-measurable set in R is in A. In this case, the set of linear spaces on which the partial expectation is defined is given by all bounded, measurable functions on the σ-blocks.

**Proposition 13** (Finitely Additive Probability on Dynkin Systems and its Partial Expectation)**.**
*Let $D_{\sigma }\subseteq 2^{\Omega }$ be a Dynkin system on the base set Ω. Let $\mu :D_{\sigma }\to \left[0,1\right]$ be a finitely additive probability defined on the Dynkin system. Then μ is in one-to-one correspondence to a partial expectation $E:{⋃}_{i\in I}B\left(\Omega ,A_{i}\right)\to R$ defined on the union of linear spaces of measurable gambles induced by all σ-blocks $A_{i}$ of $D_{\sigma }$.*


**Proof.** By Theorem A2, we can decompose the Dynkin system $D_{\sigma }$ into a set of σ-blocks ${\left\{A_{i}\right\}}_{i\in I}$. Since, $A_{i}\subseteq 2^{\Omega }$ for all i∈I, each of the σ-blocks induce a linear subspace of B(Ω), which we denote as $B(\Omega ,A_{i})$. Given a finitely additive measure $\mu :D_{\sigma }\to \left[0,1\right]$, we now define $E:{⋃}_{i\in I}B\left(\Omega ,A_{i}\right)\to R$ by
$$\begin{array}{c} {E\left(f\right):=\int fd\mu |}_{A_{i}}\;\mathrm{if}\;f\in B\left(\Omega ,A_{i}\right). \end{array}$$For every i∈I, ${E|}_{B(\Omega ,A_{i})}$ is a linear prevision in one-to-one correspondence to the finitely additive probability ${\mu |}_{A_{i}}$[2] (Theorem 3.2.2). Hence, conditions (a) and (b) in Definition 8 are met [2] (Theorem 2.8.4). For $f\in B\left(\Omega ,A_{i}\right)\cap B\left(\Omega ,A_{j}\right)$ (i≠j,i,j∈I), we know that $f\in B(\Omega ,A_{i}\cap A_{j})$ (Lemma A5); hence,
$$\begin{array}{c} {\int fd\mu |}_{A_{i}}=\int fd\mu {{|}_{A_{i}\cap A_{j}}=\int fd\mu |}_{A_{j}}. \end{array}$$Thus, *E* is well-defined and there is no other partial expectation which agrees with μ on the indicator gambles of the sets in D. □

It remains to emphasize that there are partial expectations defined on families of linear subspaces which are not induced by finitely additive probabilities on Dynkin systems. A simple example is given by a linear space which does not contain the constant gamble corresponding to the indicator gamble of the set Ω. Hence, the definition of a partial expectation is indeed a generalization of the definition of a finitely additive probability on a pre-Dynkin system.

Under the name “partially specified probabilities”, Lehrer [69] introduced a closely related notion to our partial expectation. Lehrer, however, assumed that there is by definition an underlying probability distribution over the entire base set (or better said, a σ-algebra on the base set). Hence, his partially specified probabilities are by definition extendable (see Definition 10), a fact, which he implicitly exploited by re-defining the natural extension following [2] (Lemma 3.1.3 (e)) of partially specified probabilities [70] (§3.2). Lehrer did not ask for the structure of the set of gambles with precise expectations, nor did he draw any connection to Walley’s work, nor did he link his “partially specified probabilities” to finitely additive probabilities on pre-Dynkin systems.

### 6.2. System of Precision—The Space of Gambles with Precise Expectations

In Section 3, we have shown that imprecise probabilities are precise on (pre-)Dynkin systems. The natural analogue of this *set structure of precision* is the *space of gambles with precise expectation*, which actually forms a linear subspace. The space of gambles with precise expectation is known as a set of ambiguity-free or unambiguous gambles in the finance and decision theory literature [71].

**Theorem** **7.**
*(Imprecise Expectations Are Precise on a Linear Subspace of Precise Gambles) Let B(Ω) be the linear space of bounded, real-valued functions on *Ω*. Let L:B(Ω)→R and U:B(Ω)→R be two functionals, for which all the following properties hold:*
*(a)* *Normalization: $L\left(\chi_{\Omega }\right)=U\left(\chi_{\Omega }\right)=1$.**(b)* *Conjugacy: U(f)=−L(−f) for f∈B(Ω).**(c)* *Subadditivity of U: for f,g∈B(Ω), we have U(f+g)≤U(f)+U(g).**(d)* *Superadditivity of L: for f,g∈B(Ω), we have L(f+g)≥L(f)+L(g).**(e)* *Positive Homogeneity: for α∈[0,∞) and f∈B(Ω), we have L(αf)=αL(f) and U(αf)=αU(f).**Then L and U coincide on the linear space S:={f∈B(Ω):L(f)=U(f)}⊆B(Ω), the* space of gambles with precise expectation, *which contains all constant gambles.*

**Proof.** We define
(3)$$\begin{array}{c} S:=\{f\in B(\Omega ):L(f)=U(f)\}, \end{array}$$
and show that S forms a linear subspace of B(Ω). First, let f,g∈S, then
$$\begin{array}{c} L\left(f\right)+L\left(g\right)\overset{\left(d\right)}{\leq }L\left(f+g\right)\overset{\left(★\right)}{\leq }U\left(f+g\right)\overset{\left(c\right)}{\leq }U\left(f\right)+U\left(g\right)\overset{Eq.3}{=}L\left(f\right)+L\left(g\right). \end{array}$$For (★) observe that L(f)≤U(f) for all f∈B(Ω), since
$$\begin{array}{c} L\left(f\right)+L\left(-f\right)\leq L\left(0\right)\overset{\left(a),(e\right)}{=}0\overset{\left(a),(e\right)}{=}U\left(0\right)\leq U\left(f\right)+U\left(-f\right), \end{array}$$
we have
$$\begin{array}{c} L(f)+L(-f)\leq U(f)+U(-f)\Leftrightarrow L(f)-U(f)\leq U(f)-L(f)\Leftrightarrow L(f)\leq U(f). \end{array}$$Second, let f∈S and α∈R. If α≥0, then
$$\begin{array}{c} L\left(\alpha f\right)\overset{\left(e\right)}{=}\alpha L\left(f\right)\overset{Eq.3}{=}\alpha U\left(f\right)\overset{\left(e\right)}{=}U\left(\alpha f\right). \end{array}$$Otherwise,
$$\begin{array}{c} L\left(\alpha f\right)\overset{\left(b\right)}{=}-U\left(-\alpha f\right)\overset{\left(e\right)}{=}\alpha U\left(f\right)\overset{Eq.3}{=}\alpha L\left(f\right)\overset{\left(b\right)}{=}-\alpha U\left(-f\right)\overset{\left(e\right)}{=}U\left(\alpha f\right). \end{array}$$Third, S contains all constant gambles by (a), (b) and (e). Hence, we have shown that S forms a linear subspace of B(Ω) which contains all constant gambles. □

The choice of properties for the lower and upper expectation functional is not arbitrary. We tried to resemble the properties involved in the analogous statement for lower and upper probabilities (Theorem 2). One can easily check that a lower and upper expectation with the given properties (a)–(d) forms a lower and upper probability as required in Theorem 2 if the expectation is restricted to indicator gambles. However, we added property (e), positive homogeneity.

Without the property of positive homogeneity, the resulting set of gambles with precise expectations would not form a proper linear subspace, as then one can only guarantee closedness of S under rational multiplication. The condition of positive homogeneity “fills up” the gaps with all real-scaled functions. Instead of positive homogeneity one can as well demand a continuity assumption of the lower and upper functional *L* and *U*, e.g., [2] (Property (l) Theorem 2.6.1).

It is known that the system of precision of coherent lower and upper probability is a linear space [72]. The derivation here slightly generalizes this to not necessarily coherent lower and upper functionals *L* and *U*. We emphasize that coherent previsions (see Definition 9) fulfill all of the demanded properties [2] (Theorem 2.6.1).

In Theorem 2, we did not only derive the structure of the system of precision for events, we as well showed that the restriction of the lower and upper probability to this system give us a probability defined on a (pre-)Dynkin system. In the statement here, we can define a similar restriction P:S→R with ${P:=L|}_{S}{=U|}_{S}$. However, *P* is not necessarily coherent *and* not necessarily a partial expectation. In other words, we cannot directly recover a partial expectation on subdomain of B(Ω) induced by a lower and upper expectation functional. For probabilities, this recovery was possible.

But, we can sidestep this restriction. Any pair of lower and upper expectations, as we defined them here, induce a unique lower and upper probability. The resulting finitely additive probability on the set structure of precision gives rise to a partial expectation (Proposition 12) on a set of linear subspaces contained in the space of gambles with precise expectation S of *L* and *U*.

A converse construction, however, is not possible. There is no unique lower and upper expectation functional with the given properties associated with a lower and upper probability fulfilling the axioms of Theorem 2 [2] (§2.7.3). Concluding, lower and upper expectation as defined here are not the “perfect” analogues of lower and upper probabilities.

This as well explains the mismatch between systems of precision for probabilities and expectations. A lower and upper expectation fulfills the properties of a lower and upper probability but not vice versa. Hence, only weaker statements about the system of precision are possible for probabilities. As a result, the analogue of the set structure of precision, a pre-Dynkin system, is the space of gambles with precise expectations, a *single* linear subspace. In Definition 8, however, we equated pre-Dynkin systems with *sets* of linear subspaces. In this case, a one-to-one correspondence between a finitely additive probability on a pre-Dynkin system and a generalized expectation, concretely a partial expectation, can be established. Hence, the analogy of pre-Dynkin systems and linear subspaces of gambles depends on the correspondence of probability and expectation.

### 6.3. Generalized Extendability Is Equivalent to Coherence

Partial expectations are, as we have shown, a natural generalization of finitely additive probabilities on pre-Dynkin systems. Hence, it is not far-fetched to ask for definitions of coherence and extendability again, now in the more general context. It turns out that the same story can be re-told on a more general scale: The definition of coherent probabilities (Definition 5) is in fact just the reduction of the following definition of a coherent prevision to indicator gambles.

**Definition 9** (Coherent Prevision [2] (Definition 2.5.1))**.** *Let L⊆B(Ω) be an arbitrary subset of the linear space of bounded functions. A functional $\underline{E}:L\to R$ is a* coherent lower prevision *if and only if*
$$\begin{array}{c} \underset{\omega \in \Omega }{\sup}\sum_{i=1}^{j}\left(f_{i}\left(\omega \right)-\underline{E}\left(f_{i}\right)\right)-m\left(f_{0}\left(\omega \right)-\underline{E}\left(f_{0}\right)\right)\geq 0, \end{array}$$*for non-negative n,m∈N and $f_{0},f_{1},\ldots f_{n}\in L$. If L=−L, the conjugate* coherent upper prevision *is given by $\bar{E}\left(f\right):=-\underline{E}\left(-f\right)$ for all f∈L. If, furthermore, $\underline{E}\left(f\right)=\bar{E}\left(f\right)$ for all f∈L, we call $\nu :=\underline{E}=\bar{E}$ a*linear prevision.

This definition of coherent previsions is substantiated by consistency of gamblers regarding their betting behavior on gambles with uncertain outcome, e.g., [2] (§2.3.1). Importantly, the rather opaque but general definition of coherence can be simplified greatly for coherent previsions defined on linear subspaces of B(Ω). Theorem 2.5.5 in [2] shows that coherence for lower previsions on linear subspaces can be expressed as superadditivity, positive homogeneity, and accepting sure gains (see [2] (Definition 2.3.3)).

Having introduced the notion of a coherent prevision, we now envisage the link between partial expectations and coherent previsions. We introduced extendability for finitely additive probabilities on pre-Dynkin systems as a useful property. It guarantees that the probability can “nicely” be embedded into “larger” finitely additive probability which is defined on an encompassing algebra. Hence, the analogue for partial expectations is straightforward.

**Definition 10** (General Extendability)**.***A partial expectation $E:{⋃}_{i\in I}L_{i}\to R$ is* extendable *if and only if there exists a partial expectation $E^{\prime }:B\left(\Omega \right)\to R$ such that $E^{\prime }{|}_{{⋃}_{i\in I}L_{i}}=E$.*

Interestingly, the extendability condition provided in Theorem 3 has a (more general) cousin adapted to the setting of gambles instead of events.

**Proposition 14** (Extendability Condition for Previsions)**.***(cf. [60] (Theorem 6.1).) Let ${\left\{L_{i}\right\}}_{i\in I}$ be a non-empty family of linear subspaces of B(Ω). A* partial expectation *$E:{⋃}_{i\in I}L_{i}\to R$ is extendable if and only if for every finite collection of functions $f_{1},\ldots ,f_{n}\in {\left\{L_{i}\right\}}_{i\in I}$,*
$$\begin{array}{c} \sum_{i=1}^{n}f_{i}\geq 0\Rightarrow \sum_{i=1}^{n}E\left(f_{i}\right)\geq 0. \end{array}$$

**Proof.** It seems that Theorem 6.1 [60] is equivalent to our statement. However, there is a subtlety which we want to argue here is indeed irrelevant. Extendability of a partial expectation requires the existence of a positive, *normed*, linear functional on B(Ω), whose restriction on the according linear subspaces coincides with the partial expectation. Theorem 6.1 in [60] only guarantees that a positive, linear functionals exists. But, normedness of such functional is automatically given if $\chi_{\Omega }\in L_{i}$ for some i∈I. Otherwise, we extend the partial expectation *E* to $E^{\prime }:\left\{\alpha \chi_{\Omega }:\alpha \in R\right\}{⋃}_{i\in I}L_{i}\to R$ such that
$$\begin{array}{c} E^{\prime }\left(f\right):=\left\{\begin{array}{c} \alpha \;\mathrm{if}\;f\in \{\alpha \chi_{\Omega }:\alpha \in R\}\\ E(f)\;\mathrm{otherwise}. \end{array}\right. \end{array}$$Then again, Theorem 6.1 [60] applies. □

Against the background that extendability and coherence define the same concept for finitely additive probabilities on pre-Dynkin systems, the resulting equivalence of extendability and coherence for partial expectations is of little surprise.

**Proposition 15** (Extendability is Equivalent to Coherence)**.**
*Let ${\left\{L_{i}\right\}}_{i\in I}$ be a non-empty family of linear subspaces of B(Ω). The partial expectation $E:{⋃}_{i\in I}L_{i}\to R$ is extendable if and only if E is a linear prevision, i.e., is coherent.*


**Proof.** If *E* is a linear prevision on ${⋃}_{i\in I}L_{i}$, then there is a linear prevision $E^{\prime }:B\left(\Omega \right)\to R$ such that $E^{\prime }{|}_{{⋃}_{i\in I}L_{i}}=E$ [2] (Theorem 3.4.2). Conversely, if *E* is an extendable partial expectation, then its extension is obviously a linear prevision, and hence it is coherent. The restriction of a coherent linear prevision to any subset of gambles is coherent (and linear). □

### 6.4. A Duality Theory for Previsions and Families of Linear Subspaces

In Section 5, we step by step spelled out an order relationship between the set structure of precision and credal sets, a model for (coherent) imprecise probabilities. Naturally the presented generalization begs the question whether a related relationship between credal sets and the spaces of gambles with precise expectations exists. We answer affirmatively. We redefine the credal and dual credal set function and shortly discuss its analogous properties. Again, we require a “reference measure”. In this case, it is a fixed linear prevision ψ on the space of all gambles B(Ω), which is indeed in one-to-one correspondence to a finitely additive probability measure on $2^{\Omega }$.

**Definition 11** (Generalized Credal Set Function)**.***Let *Δ* be the set of linear previsions on the Banach space B(Ω). For a fixed linear prevision ψ∈Δ, we call*$$\begin{array}{c} m_{\psi }:2^{B(\Omega )}\to 2^{\Delta },\;m_{\psi }\left(G\right):=\left\{\nu \in \Delta :\nu \left(g\right)=\psi \left(g\right),\forall g\in G\right\}, \end{array}$$*the* generalized credal set function.

**Definition 12** (Generalized Dual Credal Set Function)**.***Let *Δ* be the set of linear previsions on the Banach space B(Ω). For a fixed linear prevision ψ∈Δ, we call*$$\begin{array}{c} m_{\psi }^{\circ }:2^{\Delta }\to 2^{B(\Omega )},\;m_{\psi }^{\circ }\left(Q\right):=\left\{g\in B\left(\Omega \right):\nu \left(g\right)=\psi \left(g\right),\forall \nu \in Q\right\}, \end{array}$$*the* generalized dual credal set function.

Why can we call those functions “generalized”? Simply because any system of sets is equivalently represented as its set of indicator gambles which span their own linear space of simple gambles, i.e., linear combinations of indicator gambles.

The generalized credal set function maps, as the credal set function in Definition 6, to weak${}^{★}$-closed, convex subsets of Δ. The generalized dual credal set function, however, reveals a first subtlety. It maps to linear subspaces of B(Ω). The dual credal set function following Definition 7 mapped to pre-Dynkin systems. In Proposition 12 and Proposition 13, families of linear subspaces were the analogues of (pre-)Dynkin systems. Here, a single linear subspace is the analogue of a pre-Dynkin system. For a first step towards an explanation of this asymmetry, see Section 6.2. Finally, the pair of functions constitute a Galois connection.

**Proposition 16** (Properties of Generalized (Dual) Credal Set Function)**.**
*Let $m_{\psi }$ be a generalized credal set function and $m_{\psi }^{\circ }$ be a generalized dual credal set function. All the following properties hold:*

*(a)* 
*The generalized credal set function $m_{\psi }$ maps to weak${}^{★}$-closed, convex sets.*
*(b)* 
*The generalized dual credal set function $m_{\psi }^{\circ }$ maps to a linear subspace.*
*(c)* 
*The generalized credal set function $m_{\psi }$ and generalized dual credal set function $m_{\psi }^{\circ }$ form a Galois connection.*



**Proof.** (a) We have fixed ψ to a linear prevision. Hence, it is coherent. For any G⊆B(Ω), $m_{\psi }\left(G\right)$ is the set of all linear previsions which dominate ψ on G. Theorem 3.6.1 in [2] then states that this set is weak${}^{★}$-closed and convex.
(b)Let Q⊆Δ.**Additivity**   Let $f,g\in m_{\psi }^{\circ }\left(Q\right)$. Then, for all ν∈Δ,
$$\begin{array}{c} \nu (f+g)=\nu (f)+\nu (g)=\psi (f)+\psi (g)=\psi (f+g), \end{array}$$
i.e., $f+g\in m_{\psi }^{\circ }\left(Q\right)$.**Homogeneity**   Let $f\in m_{\psi }^{\circ }\left(Q\right)$ and α∈R. Then, for all ν∈Δ,
$$\begin{array}{c} \nu (\alpha f)=\alpha \nu (f)=\alpha \psi (f)=\psi (\alpha f), \end{array}$$
i.e., $\alpha f\in m_{\psi }^{\circ }\left(Q\right)$. For homogeneity we need the easy fact that a linear prevision is not only positive homogeneous but generally homogeneous. For this consider a linear prevision ν and any gamble f∈B(Ω) with α<0, then ν(αf)=−ν(−αf)=αν(f).(c)The two functions constitute a Galois connection (cf. Proposition 4), $G\subseteq m_{\psi }^{\circ }\left(Q\right)\Leftrightarrow Q\subseteq m_{\psi }\left(G\right)$. To this end, we show the left to right implication,
$$\begin{array}{c} \nu \in Q\Rightarrow \nu \left(g\right)=\psi \left(g\right),\forall g\in G\Rightarrow \nu \in m_{\psi }\left(G\right), \end{array}$$
and the right to left implication,
$$\begin{array}{c} g\in G\Rightarrow \nu \left(g\right)=\psi \left(g\right),\forall \nu \in Q\Rightarrow g\in m_{\psi }^{\circ }\left(Q\right). \end{array}$$
This concludes the proof. □

Galois connections possess a series of helpful properties [64] (§V.7 and V.8). For instance, they give rise to a bipolar-closure operator. A non-empty subset Q⊆Δ is bipolar-closed if and only if $Q=m_{\psi }\left(m_{\psi }^{\circ }\left(Q\right)\right)$. A non-empty subset G⊆B(Ω) is bipolar-closed if and only if $G=m_{\psi }^{\circ }\left(m_{\psi }\left(G\right)\right)$. Furthermore, Proposition 16 provides necessary conditions for bipolar-closed sets. For instance, a bipolar-closed set G⊆B(Ω) is a linear subspace. But is every such linear subspace a bipolar-closed set? No.

**Example** **16.**
*Let $\Omega_{2}:=\left\{1,2\right\}$, then $B\left(\Omega_{2}\right)=\left\{\alpha_{1}\chi_{\left\{1\right\}}+\alpha_{2}\chi_{\left\{2\right\}}:\alpha_{1},\alpha_{2}\in R\right\}$. Hence, linear functionals on $B(\Omega_{2})$ are defined via their behavior on the basis. Let $\psi (\chi_{\left\{1\right\}})=1$. Then, $\left\{\alpha_{1}\chi_{\left\{1\right\}}:\alpha_{1}\in R\right\}\subseteq B\left(\Omega_{2}\right)$ is a linear subspace, but it is, as one can easily check, not bipolar-closed, because the demand for normalization of any linear prevision ν which coincides with ψ on $\chi_{\left\{1\right\}}$ requires $\nu (\chi_{\left\{2\right\}})=0$.*


Again, it seems to be more intricate than expected to characterize bipolar-closed sets. For pre-Dynkin systems and finitely additive measures we already collected some first hints that sets of measure zero play an important role in the characterization of bipolar-closed sets. In the case of linear subspaces and linear previsions, we observe a similar “combinatorial restriction”. In order to improve understanding, let us replace $2^{\Delta }$ by $2^{\mathrm{ba}(\Omega )}$ in Definition 11, Definition 12 and Proposition 16 (The proposition still holds.), which is equivalent to stating that linear previsions are not necessarily normalized, nor positive. Then, by leveraging the Hahn–Banach-type Theorem 1.5.14 in [26], one can easily see that linearity of a subset G⊆B(Ω) is not only a necessary, but as well a sufficient condition for bipolar-closedness for those modified “credal set functions”. Thus, the restriction to actual linear previsions makes the characterization of bipolar-closed sets more complex. A compelling, more exhaustive answer still waits to be found.

Analogous to the discussion in Section 5.4, it is possible to provide a lattice duality and interpolation scheme via the generalized (dual) credal set functions. Instead of the lattice of pre-Dynkin systems (D,⊆), the interpolation is directed by the lattice of linear subspaces (L,⊆) of B(Ω). As commonly known, the lattice of linear subspaces has the two operations $L_{1}\wedge L_{2}:=L_{1}\cap L_{2}$ and $L_{1}\vee L_{2}:=\mathrm{lin}\left(L_{1}\cup L_{2}\right)$. We denote the linear span with lin. Its minimal element is the trivial zero vector linear subspace {0}. Its maximal element is the entire space of all gambles B(Ω). Due to higher generality of the here-presented (dual) credal set function, the interpolation provided is more fine-grained than for the previously given interpolation by pre-Dynkin systems. The following set containment (trivially) holds:$$\begin{array}{c} \left\{m_{\psi }\left(D\right):D\in D\right\}\subseteq \left\{m_{\psi }\left(L\right):L\in L\right\}, \end{array}$$
where $m_{\psi }\left(D\right)=m_{\psi }\left(\left\{\chi_{D}:D\in D\right\}\right)$. In other words, the lattice of pre-Dynkin systems is “contained” in the lattice of of linear subspaces. However, as for pre-Dynkin systems the interpolation via linear subspaces is improper. The reason for this is again that not every linear subspace is bipolar-closed. By restriction to linear, bipolar-closed subspaces one can clean up the setup. For details we refer to Section 5.4. We do not make explicit the detailed reiteration of the same argument here.

In summary, we confirmed our findings of Section 5 extended to previsions. The generalized dual lattice setup underlines the structural consistency between the system of precision and its corresponding imprecise probability.

## 7. Conclusions and Open Questions

In this paper, we have explicated relations between the systems of precision and imprecise probabilities (respectively expectations). First, we have shown that the system of precision forms a pre-Dynkin system (respectively, a linear subspace). This structural insight raises a series of follow-up questions: How does the system of precision of a coherent prevision relate to the set of desirable gambles of this prevision? How does the preference ordering change the set structure of precision for the corresponding beliefs? What is the role of coherence with respect to the system of precision?

Second, we defined finitely additive probabilities on pre-Dynkin systems. The equivalence of extendability and coherence of such probabilities strengthens the link between quantum probability and imprecise probability. We speculate that further insights can be obtained by exploiting this relationship. In addition, the generalization of finitely additive probabilities on pre-Dynkin systems to partial expectations directly opens the door to machine learning applications. In robust machine learning, the expected risk minimization framework is extended to more general expectation functionals. Partial expectation can, possibly after more computational investigations, deliver the desired robustness against dependencies in specific domains, such as privacy preservation, “not-missing-at-random” features, restricted data base access or multi-measurement data.

Finally, we developed a duality theory of systems of precision and imprecise probabilities (respectively expectations). A Galois connection defines a parametrized family of imprecise probabilities which follow an order structure provided by the lattice of pre-Dynkin systems (respectively the lattice of linear subspaces).

In modern statistics, especially in machine learning, probabilistic statements are increasingly tailored to individuals. Individual probabilistic statements, however, require justification. One can interpret probabilities on pre-Dynkin systems as probabilities which do not allow for such statements in the first place. One could perceive this fact as a weakness. We, in contrast, embrace its strength, when for ethical, legislative or other reasons, individualistic assignments are harmful, unjustifiable, forbidden or not desirable. We provide a first, rough interpolation scheme via the lattice duality. It demonstrates the space of adjustability of probabilistic assumptions in real-world scenarios. The involved pre-Dynkin systems are mathematical definitions of levels of group resolution. The question of how to choose such set-systems is related to the questions of intersectionality.

Several fundamental, technical questions remain open: how does the lattice duality imposed by pre-Dynkin systems or linear spaces relate to other dualities, such as convex duality, exploited in the field of imprecise probability. Can one easily characterize the bipolar-closed sets? Why is there no clear analogy between pre-Dynkin systems and linear subspaces?

We leave this collection of intriguing questions open to future work contributing to an understanding of the system of precision and the imprecise probability model.

## Figures and Tables

**Figure 1 entropy-25-01283-f001:**
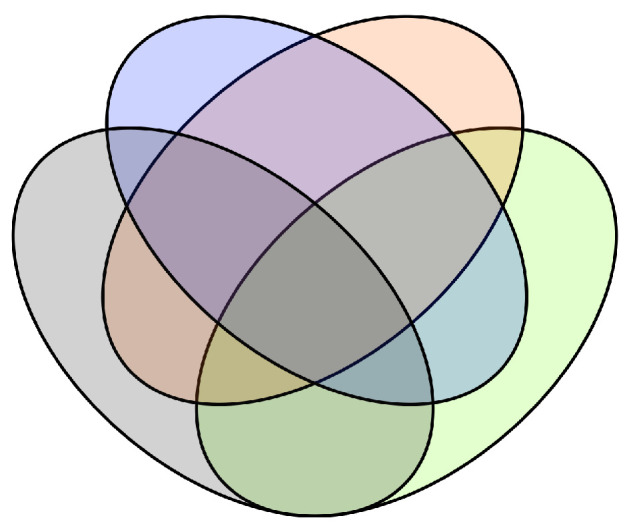
Exemplary illustration of Venn diagram for four sets (By RupertMillard, CC BY-SA 3.0).

**Figure 2 entropy-25-01283-f002:**
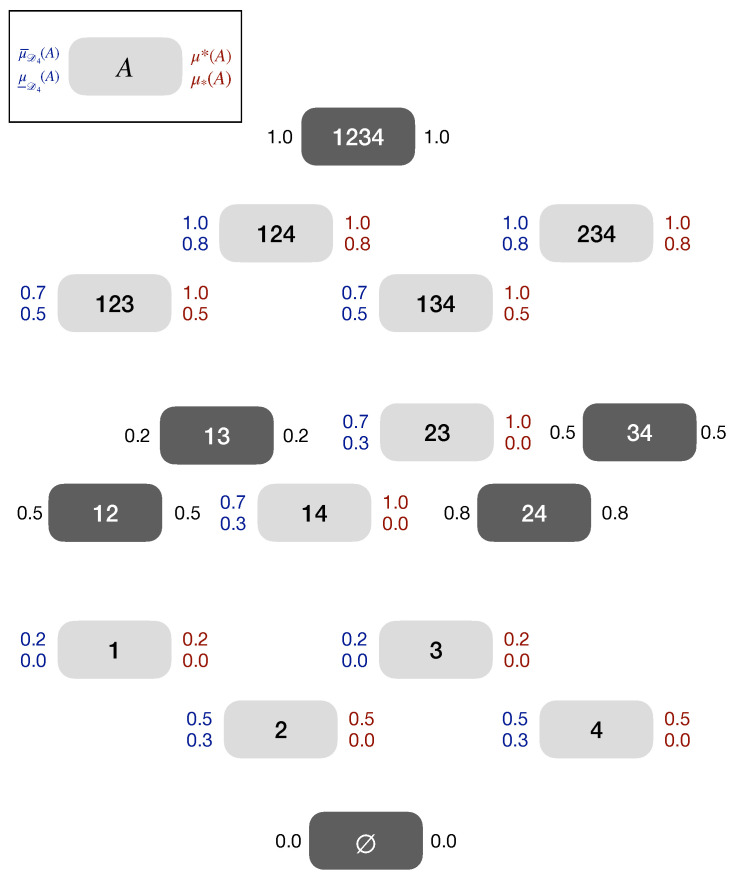
Illustration of the running example. The dark elements are contained in the pre-Dynkin system D on Ω={1,2,3,4}. The lower and upper coherent extension, respectively, the inner and outer extension, are denoted at the sides of the elements in the set system as shown in the example in the left upper corner. Elements in D possess a precise probability.

**Figure 3 entropy-25-01283-f003:**
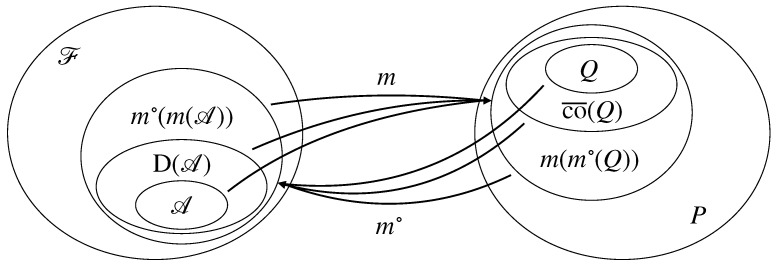
Galois connection between the lattice of pre-Dynkin systems and the set of credal sets. In the illustrated case, we have $m_{\psi }^{\circ }\left(Q\right)=m_{\psi }^{\circ }\left(m_{\psi }\left(m_{\psi }^{\circ }\left(A\right)\right)\right)$, respectively, $m_{\psi }\left(A\right)=m_{\psi }\left(m_{\psi }^{\circ }\left(m_{\psi }\left(Q\right)\right)\right)$. The set containment on both sides follows from Proposition 2, Corollary 2 and Proposition 6.

**Table 1 entropy-25-01283-t001:** Summary of important notations used.

$\Omega ,2^{\Omega }$	Base set and its power set
[n]	Set {1,…,n}
D	Pre-Dynkin system on Ω (Definition 1)
$D_{\sigma }$	Dynkin system on Ω (Definition 1)
D(A)	Pre-Dynkin hull of a set system $A\subseteq 2^{\Omega }$ (Definition 1)
μ	Finitely additive probability defined on D (Definition 3)
$\mu_{*}$, $\mu^{*}$	Inner respectively outer extension (Proposition 1)
${\underline{\mu }}_{D}$, ${\bar{\mu }}_{D}$	Lower respectively upper coherent extension (Corollary 1)
M(μ,D)	Credal set of μ on D (Corollary 1)
ν	Finitely additive probability defined on $2^{\Omega }$
ψ	Fixed, finitely additive probability defined on $2^{\Omega }$
$\chi_{A}$	Indicator function of the set A⊂Ω
Δ	Set of finitely additive probability measures on $2^{\Omega }$, set of linear previsions
$m_{\psi }:2^{2^{\Omega }}\to 2^{\Delta }$	Credal set function (Definition 6)
$m_{\psi }^{\circ }:2^{\Delta }\to 2^{2^{\Omega }}$	Dual credal set function (Definition 7)
$\bar{\mathrm{co}}\;$	Convex, Weak${}^{★}$ Closure
B(Ω)	Set of real-valued, bounded functions on Ω
ba(Ω)	Set of bounded, signed, finitely additive measures on $2^{\Omega }$
*E*	Partial Expectation (Definition 8)
S(Ω,A)	Linear space of simple gambles on the set system A
$B(\Omega ,F_{\sigma })$	Linear space of bounded, $F_{\sigma }$-measurable functions
$\underline{E}$	Coherent lower prevision (Definition 9)
$\bar{E}$	Coherent upper prevision (Definition 9)
ν	Linear prevision defined on B(Ω) (equivalent to ν above)
ψ	Fixed, linear prevision defined on B(Ω) (equivalent to ψ above)
$m_{\psi }:2^{B(\Omega )}\to 2^{\Delta }$	Generalized credal set function (Definition 11)
$m_{\psi }^{\circ }:2^{\Delta }\to 2^{B(\Omega )}$	Generalized dual credal set function (Definition 12)

## Data Availability

Not applicable.

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
