# Peer review of "Systems of Precision: Coherent Probabilities on Pre-Dynkin Systems and Coherent Previsions on Linear Subspaces [Author-notes fn1-entropy-25-01283]"

_entropy, 2023, doi:10.3390/e25091283_

Round 1

Reviewer 1 Report

In this paper, the authors investigate the problem of extending a (finitely additive) probability from a class of events to a larger class. They establish that it can be assumed without loss of generality that the original domain has the structure of a pre-Dynkin system, and then generalise a number of properties established by other authors, most prominetly Peter Walley, in the context of coherent lower probabilities and previsions.

The paper is well-written and the results are, as far as I can tell, correct. The problem is interesting and it seems to settle the issue of what is the minimal domain where coherence can be required in order to have an ‘easy’ extension. The references to the existing literature are for the most part complete, with a few minor absences. For all this, I believe that the paper has merit and deserves publication.

Nevertheless, I also have a few comments:

-Concerning the exposition, the paper goes sometimes into a lot of detail to establish things that are either trivial or well-known from the literature. This is fine for instance if we would be presenting this work in a PhD dissertation, but for a journal I think that some things can be skipped. In particular, this applies to most of what appears in the Appendix: either I would clarify their interest or shorten the paper.

-As also mentioned in the paper, the extendability problem was solved by Bruno de Finetti for arbitrary domains, and extended later by Peter Walley, using the notion of coherence. What this paper is showing basically is that there is no loss of generality in assuming that these arbitrary domains are pre-Dynkin systems. This is interesting, but it feels as if the paper is too long for the amount of novel information it gives. A similar point can be made about Section 5: the one-to-one correspondence between coherent lower previsions and weakly*-closed convex sets of probability measures was already established by Walley, so the results here will not be surprising to those familiar with imprecise probabilities. I would ask then the authors to make an effort to clarify the novelties and interest of the results established here.

-With respect to the comments at the end of Section 5.3: I was a bit surprised that the result does not hold more generally, given that you are in a finite space and then the union of all events with measure 0 will have also measure 0. Can you expand your comment a bit? I agree that for infinite spaces this would be more problematic.

-When introducing Dynkin systems in Section 2, perhaps you can mention that these are sometimes referred to as \lambda-systems in the literature (by Dynkin himself, for instance). In fact the extendability problem that is studied here could be understood better if you recall the characterisation of \sigma-algebras as Dynkin systems generated from a \pi-system, and that starting from a \pi-system is actually the problematic part of your problem.

-About Section 4.4: perhaps I missed it, but the inner and outer probability measures coincide with the lower and upper natural extensions when the original domain is an algebra, even in the imprecise case (that is, if we start from a coherent lower probability). Please check 3.1.5 in Walley. It is also interesting to note that Walley discusses the uniqueness of the extensions in Section 3.1.9.

Minor comments

-About footnote 9, this relates to the problem of defining a uniform distribution on the positive integers, that is treated thoroughly by Peter Walley in his book (see section 2.9.5 and the subsequent references). It would be interesting if you can relate your results to this problem. See also

Kadane and O’Hagan, “Using Finitely Additive Probability: Uniform Distributions on

the Natural Numbers." Journal of the American Statistical Association Vol. 90, number 430 (1995), pp. 626-631.

-Concerning the extendability problem in the precise case, it would be worth to look at the work in

S. Maass, ‘Exact functionals and their core’. Statistical papers, 43, 75-93, 2002.

 -Proof of Proposition A21: why does the weak convergence imply the first equality in the display? Shouldn’t A be a continuity set? Please clarify.

 -With respect to Appendix D.5, perhaps the following references are of interest:

 V. Krätschmer, ‘When fuzzy measures are upper envelopes of probability measures’. Fuzzy Sets and Systems, 138, 455-468, 2003.

 and also the work by Teddy Seidenfeld on conglomerability vs. finitely additive probability; see for instance

 J. Kadane, M. Schervisch and T. Seidenfeld, ‘Statistical implications of finitely additive probability’. In ‘Bayesian Inference and Decision techniques’, pages 59-76, 1986.

Typos:

-Page 26: there is a \leq duplicated in the second display of the proof.

-Page 27: ‘associated to’ (should be ‘with’).

-Page 37: there is a full stop missing at the end of the statement of Theorem A16.

Author Response

We are very thankful for the detailed feedback to our paper. In the following, we step-by-step respond to the bullet points made by the reviewer. In the attached pdf we include the revised paper with color marks:
orange - our own revision
purple - first reviewer
teal - second reviewer
olive - third reviewer

MAJOR COMMENTS

(First bullet point)
We appreciate the reviewer's feedback. However, we are very unsure to which parts this feedback refers to. Generally, we already obtained feedback asking for more details on the established background. Hence, we think that the current presentation has its merit.
To concretely look upon the appendix:
In Appendix A we present necessary proofs which, however, do not lead to new insights. We suspect that some of the statements, namely Lemma A5 (Supremum of a Chain of Algebras is an Algebra), A9 (Intersection of Simple Gamble Spaces) and A10 (Intersection of Measurable Gamble Spaces) are already present in literature. But, we were not able to identify a matching reference. For the sake of completeness (another critique by other reviewers) we decided to include the short proofs.
Appendix B, we assume, cannot be part of the critique.
Appendix C presents a, to the best of our knowledge, new investigation about distorted probabilities. We believe that the given statement is an intriguing starting point for similar investigations of systems of precision for further "types" of imprecise probability. It deserves presentation, but does not perfectly fit to the story of the main paper, hence, it is in the appendix.
Appendix D focuses on countably additive probability measures. The statements here are, as far as we know, new. This statements \emph{do} reiterate arguments given in the main paper. But, as \eg shown in [Statistical Implications of Finitely Additive Probability, Teddy Seidenfeld 1999], the subtle difference to finitely additive probabilities can lead to surprisingly different results. This section's purpose is to reestablish the given arguments for the audience originating from a classical probability-theoretical perspective, as arguable most people in \eg Machine Learning.
Appendix E might be of little interest, but highlights a neat connection of probability theory of generalised set structures to probabilities on logical structures. We are convinced that this short section might spark interest and embed our work in the web of generalised probability theories.

(Second bullet point)
We do think that our contribution consists of more than the statement: without loss of generality pre-Dynkin-systems can replace arbitrary domains in the context of extendability. For instance, we show the simple but insightful statement that the system of precision of imprecise probabilities are pre-Dynkin-systems (Theorem 10). And, Section 5 establishes a different one-to-one correspondence than the reviewer alludes to. We present a duality of systems of precision and credal sets. The mapping between credal sets and lower (coherent) previsions is not part of our presentation (because, as the reviewer noted, this is a central element of Walley's work). The duality spelled out in the paper, however, is not in Walley's book.

(Third bullet point)
This comment links into an interesting direction: what is the influence of the measure zero sets with respect to the reference measure on the bipolar-closed sets? A direct generalization as you pointed out, however, seems, at least to us, not immediately available. We now provide another example (Example 38) to emphasize that even in the finite setting the Pre-Dynkin-Hull operator and the Bipolar-Closure are not necessarily equivalent, when there are non-empty measure zero sets. It is a common frustration [Gian-Carlo Rota, Twelve problems in probability no one likes to bring up, 2001] that measure zero sets "should" be ignorable, but are not. Our results on the bipolar-closure are yet another example of this phenomenon.

(Fourth bullet point)
This is indeed true. We hint to \lambda-systems on the first page footnote 2, which refers to Appendix B.
We agree with the referee's comment that starting from \pi-systems is the critical part. We tried to emphasize this point at several places:
(i) page 5: "In classical probability theory, Dynkin-systems appear as a technical object required for the measure-theoretic link between cumulative distribution functions and probability measures (cf. [ 23 , Proof of Lemma 1.6])."
(ii) page 7: "Interestingly, the assumption of arbitrary compatibility is fundamental to most parts of probability theory. \sigma-algebras, the domain of probability measures, are exactly those Dynkin-systems in which all events are compatible with all others [34, Theorem 2.1]."
(iii) In Appendix A we introduce "compatibility structures" which are to Dynkin-systems what \pi-systems are to \sigma-algebras.

We avoid \pi-systems in the main text since we tried to keep the number of fundamental objects as small as possible. \pi-Systems would not (significantly) contribute to other statements in this work.

(Fifth bullet point)
Thank you for that remark. We were aware of these results but missed to mention them in the text. We added the statement in Section 4.4 .

MINOR COMMENTS

(First bullet point)
We are aware of that link. For this reason we pointed towards the exhaustive work by Schurz and Leitgeb [G. Schurz , H. Leitgeb; Finitistic and Frequentistic Approximation of Probability Measures with or without σ-Additivity, 2008]. For closer discussions between (uniform) distributions on the integers and pre-Dynkin-systems see this work.

(Second bullet point)
We are not entirely sure how we should relate [S. Maass, Exact Functionals and their core, 2002] to our work. The extendability problem is, as we show, equivalent to the coherence problem (coherence understood as put forward by Walley). Maass introduces "exactifiability", which seems to be very close to coherence (Proposition 10.b). Coherent functionals are normed, exact functionals [Lower Previsions, M. Troffaes, G. De Cooman, p 220]. Hence, the extendability problem can be seen as a special case of "exactifiability".

(Third bullet point)
Thank you for carefully reading this proof. The first equality just holds in the case that A would be a continuity set. Since we require the statement to hold for all measurable sets in the Dynkin-system we now simply strengthened the topology. Instead of the weak topology we introduce the topology induced by the total variation norm (TV). Then set wise convergence of TV-converging sequences of measures can be guaranteed.

(Fourth bullet point)
Thank you for the references. We were aware of the second work, but the first work is new to us. 

Thanks for catching the Typos!

Reviewer 2 Report

This is a good paper about partially specified probability distributions which is sound and contains original results. The paper is formally precise and rigorous.  My only concern is that it can difficult to read and grasp the overall contributions. In this sense, I have some minor recommendations suggestions:

1 Though the paper has some running examples, I believe that the paper needs more of them (there is only one at the end of Section 6) and to extend the existing ones with more cases and discussion. This can help to read the paper.

2. Sometimes the language is of high level and you can not follow the comments if you are not an expert in the topic. This is particularly relevant in the relationship with quantum probability which is a central point of the paper, but it misses a mathematical presentation. For example, I am completely lost in the discussion of Subsection 4.2.1

3. The paper starts with sigma-additive probabilities as well as only additive probabilities, but then it is after Subsection 4.2 when it concentrates on only additive probabilities. For a shake of simplicity, I would concentrate the full paper on additive probabilities and Pre-Dynkin systems, as I believe that this is the most relevant case. The sigma additivity can be left for an Appendix, or even better for further papers. 

4.In page 2, you say that fuzzy logic lacks of operational definition,  I believe that this is a too strong assessment. There is a huge literature about fuzzy logic, and some papers give operational interpretations. I believe that it is better to avoid this kind of absolute statement.

5. Page 5, some more detail about the weak topology can be useful to understand the paper if you are not familiar with it (what are the open sets in this case).

6. Page 6. You can extend a little more the cases of the natural density and the marginal scenarios, perhaps giving examples.

7. Definition 9: you talk about a pre-Dynkin system and sigma additivity.

8. Theorem 10 is OK, though I earlier recommended stating only the first part. It is stated for general subadditive and superadditive upper and lower probabilities. As I am from the first of imprecise probability, for me only coherence makes sense for a pair of upper and lower probabilities. I would have stated it for coherent probabilities, though I recognize that the current result is more general.

9. Page 11- For me the inner and outer extension is only a mathematical concept and only the coherent extension (which as it is shown is more informative) makes sense. Some comments or discussion about this could be useful.

10. I like Theorem 19 in which extendability equals coherence (and not surprising at all).

11. Section 4.2.2 is another case of discussion without precise statements that is not easy to understand, though I am familiar with the work of Casanova et al.

12. Section 5 needs more examples with a discussion about their intuitions, in another case, though you can follow the mathematics it is not simple to grasp the meaning.  Sometimes you forget that there is an underlying probability in the process. The discussion about what it represents should be also added.

13. The beginning of Section 5.3.1 is a bit vague. When you say 'Bipolar-closed subsets \Delta require further investigation' Is that investigation what follows or it is something for future research? You can be more specific about the content of the subsection.

14. Page 27. In the discussion at the top the page I am lost several times. For example, when it is said that the restriction L|S = U|S is not necessarily a partial expectation .... or when making reference to Theorem 10 [3, 2.7.3]  But in general, I believe that all the content should be made easier to read.

Author Response

We are very thankful for the detailed feedback to our paper. In the following, we step-by-step respond to the bullet points made by the reviewer. In the attached pdf we include the revised paper with color marks:
orange - our own revision
purple - first reviewer
teal - second reviewer
olive - third reviewer

1.
We are a bit sceptical that more and longer examples will contribute to a better understanding of the paper. The subject itself is abstract and we hope to get a glimpse of an intuition transferred by the given examples. In particular, larger examples, i.e. not working on toy base sets of 4 elements, rapidly get very messy. Section 6 lacks more examples exactly because of this reason.
Last but not least, for the sake of brevity (the paper already is relatively long + other reviewers complained it was indeed too long) we refrain from giving further examples.

2.
We can, as the reviewer, definitely identify Section 4.2.1 as being high level. We explicitly aimed for this level of abstraction. A more detailed introduction to compatibility and contextuality is out of the scope of this work. 
Other reviewers asked for shortening the work. Hence, we see the arguments counterweighting each other here.
To help the reader we added the statement that the given discussion Section 4.2.1 is particularly aimed at persons familiar with quantum probability concepts such as contextuality and compatibility.

3.
We potentially irritated the reviewer by introducing countably additive probabilities before introducing finitely additive probabilities. We now turned around the definition (Definition 9) to emphasize that our main workhorse are finitely additive probabilities. Countably additive probabilities do only (as part of a mathematical statement) occur in Theorem 10. All further discussions have been shifted to the appendix. We think that it would be unreasonable to split up Theorem 10 for the only reason to ban countably additive probabilities from the main paper. The result is requires almost identical tools for countably as for finitely additive probabilities.
Furthermore, we want to emphasize that finitely additive probabilities are, in some communities, still treated as exotic. We hope to engage scholars in those communities to think about fundamental questions such as the domain of probabilities by explicitly referring to countably additive probabilities.

4.
In fact, this statement is rather strong and, it is not ours. In his paper "The anatomy of the squizzel: The role of operational definitions in representing uncertainty", 2004, Roger Cooke argues for this statement. We find his argumentation appealing. Nevertheless, we smoothed the statement and pointed more explicitly towards Cooke's paper.

5.
We indeed need weak(^*) topologies as a technical tool to work on firm ground, but this topology is not central to our argumentations. Nevertheless, we now redirect the reader to a solid introduction to weak(^*) topologies by Eric Schechter in "Handbook of Analysis and its Foundations", 1997.

6.
Natural density and marginal scenarios are indeed interesting subcases of our more general theory of probability on pre-Dynkin-systems. Particularly, marginal scenarios form a well-studied class of problems which we embed in a more general framework via pre-Dynkin-systems (cf. Example 4.2 in [An Extension of Classical Measure Theory, 1984, S. Gudder]).
A more exhaustive introduction would be, as we think, beyond the scope of this work. Nevertheless, we have rewritten Section 4.2.2 "Extendability and Marginal Problem" to convey a more informal intuition. Additionally, we want to emphasize that we already point towards existing works in marginals scenarios [Verteilungsfunktionen mit gegebenen Marginalverteilungen, 1964, H. G. Kellerer][Consistent families of measures and their extensions, 1962, N.N. Vorob'ev] and natural densities [Finitistic and Frequentistic Approximation of Probability Measures with or without σ-Additivity, 2008, G. Schurz and H. Leitgeb].

7.
Countably additive probabilities can be defined on pre-Dynkin-system. To this end one has to, and indeed we forgot in the former version, demand the countable disjoint union of elements to be in the pre-Dynkin-system. We corrected the definition accordingly.

8.
For this reason we emphasize below the proof that coherent lower and upper probabilities fulfill the demanded properties.

9.
The coherent extension is as well only a mathematical concept, just carrying more interpretational value. The interpretational value is in parts build on the fact that coherent extensions do fulfill more (helpful) properties. We do emphasize the ``weakness'' of the inner and outer measure construction at several points (Section 4.1).

10.
We are pleased to hear that. :)

11.
We reworked the section and illustrated it via a rough non-rigorous example. We hope to make the statement clearer.

12.
The section already contains several examples: 24, 26, 33, 34, 40 (former 39), 42 (former 41) and we added an example 38. The examples are tailored to our needs to show specific properties of the mappings presented in this section, but there is no direct interpretative value of the given toy examples.
Regarding the underlying probability, we appreciate the feedback and have made the notation more explicit by rewriting the credal and dual credal set function with a subscript $\phi$ referring to the reference measure, which, as the reviewer stated, is always present.

13.
Thank you for the remark, we changed the statement accordingly. We meant future investigations.

14.
We restructured section to guide the reader more easily through this discussion.

Reviewer 3 Report

The paper proposes an analysis of probabilities defined on set structures called Pre-Dynkin-Systems. The analysis is related to the theory of imprecise probabilities. One of the main motivations for so defined probabilities is that the parts where imprecise probabilities are precise are Pre-Dynkin-Systems. A dual system of maps mapping from collections of subsets to credal sets where the probabilities equal to a reference probability and vice-versa. It is shown that the composition of these maps forms Pre-Dynkin-Systems. Moreover, fixed points of the composed maps are partially characterized. The results are also extended from probabilities to expectations on unions of linear space.

My general opinion is that the paper proposes some interesting results and would recommend it to be accepted to publication in Entropy after some minor corrections listed below.

1.      Regarding Theorem 10, I recommend incorporating (1) as a component of the theorem itself. The reason is that (1) provides a stronger statement than what is currently claimed in the theorem. Presently, the theorem states the existence of a set $\mathcal{D}$ with a particular property, but (1) actually provides the precise form of this set. I would suggest applying the same approach to Theorem 47 as well.

2.      In Theorem 19, the paper asserts that a coherent additive probability results in a non-empty credal set. However, this claim appears to be a specific instance of a more general result known from Walley's theory, wherein a coherent lower-upper probability pair leads to a non-empty credal set, with the bonds formed as its upper and lower envelope. I believe this relationship warrants further discussion in the paper. Additionally, there is another property, namely "avoiding sure loss," which seems to serve the purpose of extendability. Surprisingly, this property is not addressed in the paper. It would be beneficial for the authors to include a discussion on this matter. Specifically, it would be interesting to explore whether avoiding sure loss is equivalent to coherence within the context of their work.

3.      The paper introduces a credal set function that relies on a specific finitely additive probability. However, it lacks an explanation regarding the significance of this probability measure and how its selection influences the associated structures. It would be beneficial to include a discussion on the role of the reference probability, providing insights into its implications. This additional explanation would enhance the reader's understanding of the paper's key concepts.

Typos:

1.      L. 106: remove ‘with’

2.      Proof of Theorem 37: equation before l. 689, I think you mean $\psi(A)$ instead of $\mu(A)$.

3.  L. 683: remove ‘sets’

4.  Proof of Theorem 47, second equation, first inequality: remove one $\le$

A few typos are within general comments

Author Response

We are very thankful for the detailed feedback to our paper. In the following, we step-by-step respond to the bullet points made by the reviewer. In the attached pdf we include the revised paper with color marks:
orange - our own revision
purple - first reviewer
teal - second reviewer
olive - third reviewer

1.
Thank you for this remark. This is indeed a good idea. We changed the statements accordingly.

2.
Yes, Theorem 19 shows that the credal set of an extendable probability on a pre-Dynkin-system is non-empty. In fact, one can use credal sets to recover the statement about equivalence of extendability and coherence --  we added that statement. And we have now alluded to "avoiding sure loss", which is indeed in our case simply equivalent to coherence (see Theorem 4.12 in [Troffaes, De Cooman, 2014, Lower Previsions]). Thanks for bringing up this missing discussion.

3.
This is indeed an important "hidden" aspect of our duality. The choice of the reference measure particularly influences the measure zero sets, which play a major role in the characterization of bipolar-closed sets. We added some short statements and particularly changed notation to emphasize the dependence of the (dual) credal set function on the reference measure.

Thank you for catching the typos!

Round 2

Reviewer 1 Report

I have revised the new version of the paper, and checked their replies to my initial review. While I do not agree with some of them and think that the paper could be improved with a shorter exposition, I respect their opinion and I am fine with having this version accepted. 

Reviewer 2 Report

Though I would have liked to see more examples and I am not irritated by introducing sigma additive probabilities, I believe that the paper can be accepted as it is.